# LANGUAGE CONDITIONED EQUIVARIANT GRASP

## ABSTRACT

The ability to control robots with simple natural language instructions enables non-experts to employ robots as general tools and has long been a goal in robot learning. In this paper, we examine the problem of training a robotic grasping policy conditioned on language instructions. This is inherently challenging since efficient manipulation policy learning often exploits symmetry and geometry in the task, but it is unclear how to incorporate language into such a framework. In this work, we present **L**anguage-conditioned **E**quivariant **G**rasp (**LEG**), which leverages the $SE(2)$ symmetries of language-conditioned robotic grasping by mapping the language instruction to an $SO(2)$-steerable kernel. We demonstrate the sample efficiency and performance of this method on the Language-Grasp Benchmark which includes 10 different language-conditioned grasping tasks and evaluate it on a real robot.

## 1 INTRODUCTION

Vision Language models (VLMs) have demonstrated promising performance on a variety of tasks from image captioning to action recognition. Substantial progress has been made in adapting VLMs to command robots to follow language instructions and reach visual goals (Chen et al., 2023; Driess et al., 2023). For instance, CLIPort (Shridhar et al., 2022) combines the semantic understanding of CLIP (Radford et al., 2021) with the spatial action prediction of Transporter (Zeng et al., 2021); VIMA (Jiang et al., 2022), PerActor (Shridhar et al., 2023) and RVT (Goyal et al., 2023) fuse visuals and textual tokens to learn a language conditioned multi-task policy with the Transformer (Vaswani et al., 2017). However, directly interleaving language features with image features breaks the geometric symmetries underlying the optimal policy. As a result, these methods require a large number of demonstrations. For example, VIMA cannot learn a non-trivial policy without at least $10^4$ demonstrations. However, language-conditioned tasks also have symmetries. As illustrated in Figure 1, if

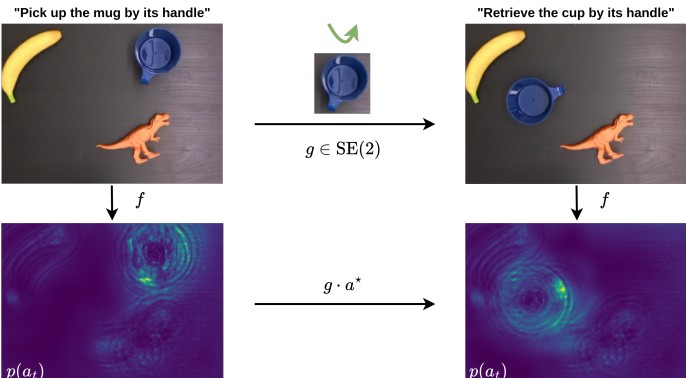

Figure 1: **Illustration of the symmetry of language-conditioned grasp.** Given the language instruction and the observation, the left subfigure highlights the action $a^*$ that could grasp the mug by its handle. A consistent grasp function should satisfy that if the language goal is unchanged but there is a transformation on the target object, the highlighted action should transform accordingly, as shown in the right subfigure.

the robot learned how to grasp a mug by the handle in the left image given a goal expressed via language, that policy should generalize to a transformed mug in the right image automatically. To leverage the symmetries of language-conditioned manipulation tasks, we use the language instruction to generate a steerable kernel (Cohen & Welling, 2017) and realize the equivariant policy, and achieve a significant improvement in sampling efficiency.

Grasping has a long history in robotics due to the needs of various applications. Previous grasping methods (Ten Pas et al., 2017; Huang et al., 2023; Mahler et al., 2017) pick every object presented in the workspace. They need a segmentation mask or other prior knowledge to grasp the object of interest. Furthermore, when it comes to grasping the target object by the specific part, e.g., *"pick the mug by its handle", "pick the fork by the top",* most pretrained vision models fail to highlight the specific proportion of the object since the fine-grain object parts are underrepresented in their training dataset. Compared to the previous methods, we present an end-to-end model that directly takes language instructions and outputs the grasp distribution over the entire action space. With 5 sets of human demonstrations, our method achieves a 90% success rate of picking the target object by the specific part instructed by the language goal on the real robot. Our method also generalizes to novel object instances.

Given that there are few ways to evaluate the performance of language-conditioned grasping, we design the *Language-Grasp benchmark* that includes 10 different tasks from picking fruits to picking captioned bottles; from picking colored toys to picking the object by the part. It inherits the Gym-like API based on Ravens (Zeng et al., 2021). Each task contains i). a scripted oracle that provides expert demonstrations and ii). a reward function that provides credit if the language goal is achieved.

Our contribution can be summarized as 1) We analyze the symmetry of language-conditioned grasping and propose a general framework to leverage it. 2) We present a novel Learning-from-Demonstration (LfD) language-conditioned grasp network, including a novel dynamic kernel generator that can map language instructions to steerable kernels with rotational symmetry. 3) We design a new benchmark that includes 10 language-conditioned grasping tasks and also provides expert demonstrations. 4) We demonstrate our inductive bias leads to high sample efficiency and grasping performance in both simulated and real-world robot experiments.

## 2 RELATED WORK

### 2.1 LANGUAGE CONDITIONED MANIPULATION POLICY

Recent breakthroughs in Natural Language Processing (NLP) and computer vision, such as Large Language Models (LLM) (Devlin et al., 2018; Brown et al., 2020; OpenAI, 2023) and Large Vision-Language Models (VLM) (Driess et al., 2023; Chen et al., 2023; Liu et al., 2023), have demonstrated impressive reasoning abilities revealing their potentials for robotic tasks. The ways to leverage the power of pretrained models for robotics can be roughly divided into two categories. The first category is using LLMs as high-level planners (Liang et al., 2023; Ahn et al., 2022; Lin et al., 2023) which assumes that the robot is equipped with low-level action skills. However, there is a gap between high-level reasoning to low-level robot actions since task-related robot data is underrepresented in their training data. The second category is to use the features from pretrained models for end-to-end training on robot data (Shridhar et al., 2022; Brohan et al., 2022; Shridhar et al., 2023; Goyal et al., 2023; Sundaresan et al., 2023; Brohan et al., 2023) (Rashid et al., 2023; Tang et al., 2023). Shridhar et al. (2022) train a multi-task agent by using features from CLIP (Radford et al., 2021) that aligns images and language during pretraining. Stepputtis et al. (2020) took GloVe word embeddings and pretrained features from Fast-RCNN to realize a language-conditioned policy. Chen et al. (2021); Shao et al. (2021) use a pretrained ResNet and a text encoder, and train the model with more than 10k samples. However, directly combining pretrained features with image features did not leverage the geometric symmetries underlying robot tasks. As a result, a copious number of robot data is required to train these models. For instance, Stepputtis et al. (2020) needs 30,000 datapoints to achieve 94% picking success rate. Jiang et al. (2022) requires 10,000 demos to achieve 80% success rate in its Level-1 tasks. Tziafas et al. (2023) uses 75k training samples to train the model. Instead, our proposed method shapes pretrained features to a steerable kernel to leverage the symmetries of language-conditioned tasks and achieve better performance with a small number of data.

## 2.2 GRASP DETECTION

This is much work in grasp detection, i.e., detecting the position and orientation of grasps based on visual observation. Key early examples are DexNet (Mahler et al., 2017) and GPD (Ten Pas et al., 2017). Most recent grasping algorithms (Morrison et al., 2018; Mousavian et al., 2019; Breyer et al., 2020; Huang et al., 2023) (Simeonov et al., 2022; Xu et al., 2023) focus on grasping the object cluttered on the table but few of them explore grasping the specific object following language goals. Compared to previous grasping methods, our method directly takes simple language instructions along with visual input and generates the action distribution of all grasps in a (planar) scene. It can not only grasp the target object successfully but also satisfy the need of grasping by the specific part instructed by the language goal.

## 2.3 EQUIVARIANT ROBOT LEARNING

Recent work using equivariant models (Wang et al., 2022d;c;a; Zhu et al., 2022; Park et al., 2022; Zhao et al., 2022; 2023b;a; Huang et al., 2022; 2023; Jia et al., 2023) have illustrated that symmetry is a vital prior for sample efficiency and spatial generalization in robotics. Although neural networks can learn an equivariance property via data augmentation (Zeng et al., 2021; Zhong et al., 2020; Krizhevsky et al., 2012; Laskin et al., 2020), symmetry-constrained neural networks often show faster convergence and more robust performance (Wang et al., 2022d; Huang et al., 2022; Jia et al., 2023). However, the aforementioned papers mainly use discrete groups, e.g. cyclic group or dihedral group, and consider the global rotation of the entire observation. In this paper, we propose a new way to map language instructions to steerable kernels and satisfy the local symmetry when an object instead of the entire observation rotates with a continuous $\mathrm{SO}(2)$ group.

## 2.4 BENCHMARKS IN ROBOTIC MANIPULATION

There are various benchmarks for robotic manipulation tasks built on different simulators (Rohmer et al., 2013; Coumans & Bai, 2016; Makoviychuk et al., 2021). Zeng et al. (2021); Wang et al. (2022b); James et al. (2020); Mu et al. (2021); Gu et al. (2023); Jiang et al. (2022); Mees et al. (2022) mainly cover pick and place tasks with simple shaped objects and a language template is also provided to describe the goal of the task. However, there is no existing benchmark designed specifically for language-conditioned grasping on various shaped, captioned, and patterned objects. In this paper, we propose *Language-Grasp Benchmark* due to the problem's significant interest in real-world applications. It includes 10 tasks from picking the bottle with a specific caption to picking an object by a specific part. We also introduce a set of metrics to measure the performance of language-conditioned grasping. To the best of our knowledge, our designed benchmark is the first to center on the specific problems of language-conditioned grasping.

## 3 METHOD

### 3.1 PROBLEM STATEMENT

This paper focuses on behavior cloning for the 2D language-conditioned robotic grasping problem. Given a set of demonstrations that contains observation-language-action tuples $(o_t, \ell_t, a_t)$, the objective is to learn a policy $p(a_t|o_t, \ell_t)$, where $a_t \in \mathrm{SE}(2)$ denote the pose of end-effector. The visual observation $o_t$ is a top-down orthographic RGB-D reconstruction of the scene that contains several objects. Since each pixel corresponds to a point in 3D space, the action $a_t$ is parameterized in terms of $\mathrm{SE}(2)$ coordinates $(u, v, \theta)$, where $u, v$ denote the pixel coordinates of the gripper position and $\theta$ denotes the gripper orientation. The gripper orientation distribution is encoded as the N-channel feature above each pixel. Each channel corresponds to a $\frac{2\pi}{n}$ rotation angle. The language instruction $\ell_t$ specifies the current-step instruction, e.g., "pick the block with a fire logo", "grasp the scissors by its handle".

### 3.2 SE(2)-EQUIVARIANT LANGUAGE-CONDITIONED GRASP

We first analyze the symmetry underlying the language-conditioned grasp and present LEG to realize the symmetry.

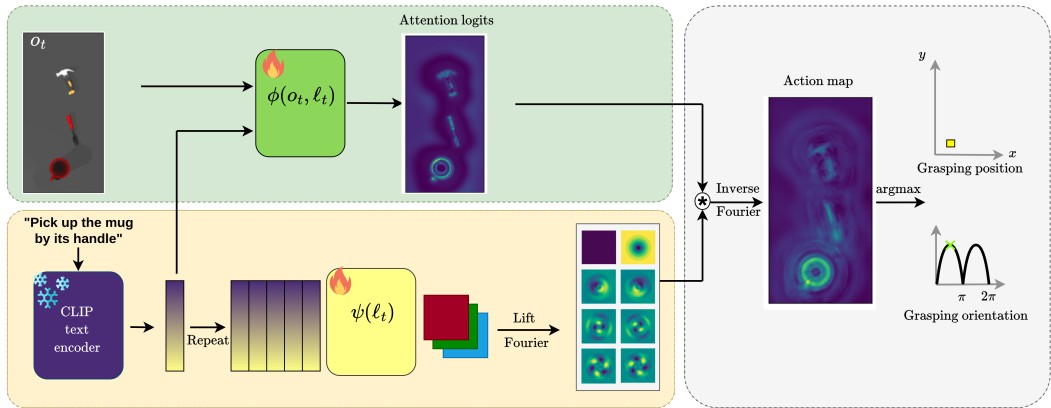

Figure 2: **Architecture of Language-conditioned Equivariant Grasp (LEG).** Our top branch (green) processes that observation and language embedding to output pixel-wise attention logits. Out bottom branch (yellow) takes the expanded language embeddings and outputs a language-conditioned steerable kernel. The grasp action is evaluated by cross-correlation between the dynamic kernel and the attention logits in Fourier space.

### 3.2.1 ANALYZING THE SYMMETRY OF LANGUAGE-CONDITIONED GRASP

Our grasping model $f$ takes as the input $o_t$ and $\ell_t$ and outputs the SE(2) pose distribution over the action space

$$f(o_t, \ell_t) = p(a_t | o_t, \ell_t) \tag{1}$$

The best pick pose can be queried by evaluating $a_t^\star = \arg\max p(a_t)$. Assume the observation $o_t$ contains a set of $m$ objects $\mathcal{B}_t = \{b_i\}_{i=1}^m$ on the workspace and denote the object $b^\ell$ as the goal object instructed by the language instruction $\ell_t$, the symmetry underlying $f$ can be stated as

$$\arg\max f(o_t^{g \cdot b^\ell}, \ell_t) = g \cdot \arg\max f(b^\ell, \ell_t) \tag{2}$$

where $o_t^{g \cdot b^\ell}$ denotes that the target object instructed by the language instructed is transformed by $g \in \text{SE}(2)$ regardless of any transformation of other objects. Equation 2 claims that if there is transformation $g \in \text{SE}(2)$ on the object $b^\ell$, the best action $a_t^\star$ to grasp the instructed object should be transformed to $g \cdot a_t^\star$.

### 3.2.2 LEVERAGING THE SYMMETRY

**Network Architecture.** To leverage the symmetry of the language-conditioned grasp, we propose Language-conditioned Equivariant Grasp (**LEG**) Network as shown in Figure 2. The framework has two branches, the top branch (shaded in green) and the bottom branch (shaded in yellow), both encoded with neural networks.

The top branch is designed to take both the observation $o_t$ and the current step language instruction $\ell_t$ as input and output the dense pixel-wise feature map. The language instruction $\ell_t$ is first tokenized and encoded to the language embedding with pretrained language models, e.g., the CLIP text encoder. Then, the language embedding and the visual observation are processed by the vision-language model $\phi$ to generate a pixel-wise dense feature map. Our entire bottom branch is denoted as $\kappa$ and takes the same language embedding and outputs a steerable kernel $\kappa(\ell_t)$ that is conditioned on the language instruction. The grasp pose distribution is calculated as the cross-correlation between $\phi(o_t, \ell_t)$ and $\kappa(\ell_t)$

$$f(o_t, \ell_t) = \kappa(\ell_t) * \phi(o_t, \ell_t) \tag{3}$$

**Vision Language Encoder: $\phi(o_t, \ell_t)$.** There are several choices of $\phi$ to encode the observation and the language instruction. We present two options here. The first one is to train from scratch.

Specifically, we use the language encoder from CLIP to generate the language embedding of the instruction and project it to a low-dimension feature with a linear layer. The observation $o_t$ is encoded by a U-Net (Ronneberger et al., 2015) and the projected language feature is attached to each pixel in the bottleneck layer. A detailed description of this $\phi$ network is described in Appendix A.3. The second option is to use the pretrained vision-language model. To be specific, we adopt the visual-language encoder from Cliport (Shridhar et al., 2022). This encoder has two networks. The *semantic network* is built on top of the pretrained CLIP (Radford et al., 2021) visual encoder and its language encoder to process the RGB image[1] of the observation as well as the language instruction. The spatial network is a Fully Convolutional Network (FCN) to process the RGB-D image only. Then, element-wise products are conducted to fuse the semantic dense feature maps and the spatial dense feature maps in the decoding process.

**Language-conditioned Steerable Kernel: $\kappa(\ell_t)$.** Our bottom branch generates a dynamic steerable kernel conditioned on the language instruction. It essentially incorporates the language elements into the geometric setting in a way that preserves the symmetry of the geometric features. We first project the same language embedding from a pretrained CLIP text encoder to a low-dimension feature that is then repeated and tiled to a 2D tensor. The network $\psi$ processes the 2D tensor and generates the dense feature map, where $\phi$ can be either a Fully Convolutional Network or a Vision Transformer. We lift $\psi(\ell_t)$ with a finite number of rotations $\{g_i \,|\, g_i \in C_n\}$ to generate a stack of rotated feature maps, where $C_n$ is a cyclic rotation group $C_n = \{\mathrm{Rot}_\theta : \theta \in \{\frac{2\pi i}{n} | 0 \le i < n\}\}$

$$\mathcal{L}[\psi(\ell_t)] = \{g_1 \cdot \psi(\ell_t), g_2 \cdot \psi(\ell_t) \cdots, g_n \cdot \psi(\ell_t)\} \tag{4}$$

Here $\mathcal{L}$ denotes the lift operation which outputs a stack of orbit-traversing signals above each pixel. This operation actually generates a steerable kernel that takes the trivial-type input and maps it to a regular-type out. This regular-type steerable is limited to discrete rotations of $C_n$. To model the $\mathrm{SO}(2)$ distribution, we apply the Fourier transform to the channel feature and generate the irreducible steerable kernel. Intuitively, the $\mathrm{SO}(2)$ signal above each pixel is represented by the coefficients of the basis functions. Detailed description can be found in Appendix A.2.

**Proposition 1** *if $\kappa(\ell_t)$ is a steerable kernel, it approximately satisfies the symmetry stated in Equation 2.*

Intuitively, if $\phi$ is an identity mapping, the cross-correlation between a steerable kernel and the $o_t$ captures the exact symmetry. That is any transformed $b^l$ will be cross-correlated at one pixel location with the steerable kernel. Detailed proof of Proposition 1 can be found in Appendix A.2.

**Bilateral Symmetry and Inverse Fourier Transformation.** The language-conditioned steerable kernel $\kappa(\ell_t)$ is cross-correlated with the dense feature map $\phi(o_t, \ell_t)$ and the outputted feature is in the shape of $\mathbb{R}^{k \times H \times W}$, where $k$ is the number of truncated frequency and $H \times W$ denote the spatial dimension. The $k$-dimension channel feature above each pixel models the $\mathrm{SO}(2)$ distribution of the gripper orientation in Fourier space. A key observation in planar picking is that, for many robots, the gripper is bilaterally symmetric, i.e., grasp outcome is invariant when the gripper is rotated by $\pi$. We can encode this additional symmetry to reduce redundancy by only keeping the signal of even frequencies that are periodic to $\pi$. Finally, a number of $n$ rotations is sampled for each pixel location with Inverse Fourier Transformation, and the best action $a^*$ is calculated by querying the $\arg\max$ over the spatial and channel dimension.

## 4 EXPERIMENTS

### 4.1 LANGUAGE-GRASP BENCHMARK

Since few benchmarks exist to effectively evaluate language-conditioned grasps, we designed a simulation environment in Pybullet (Coumans & Bai, 2016) for training and testing 2D language-conditioned grasps with a parallel-jaw Franka Gripper. It inherits the Gym-like API based on Ravens-10 (Zeng et al., 2021). Each task contains i) a scripted oracle that provides expert demonstrations and ii) a reward function that provides credit if the language goal is achieved. The observation $o_t \in \mathbb{R}^{320 \times 160 \times 4}$ is an orthographic projection from captured point clouds from three RGB-D

---

[1]CLIP is trained with RGB images and cannot handle the depth channel directly.

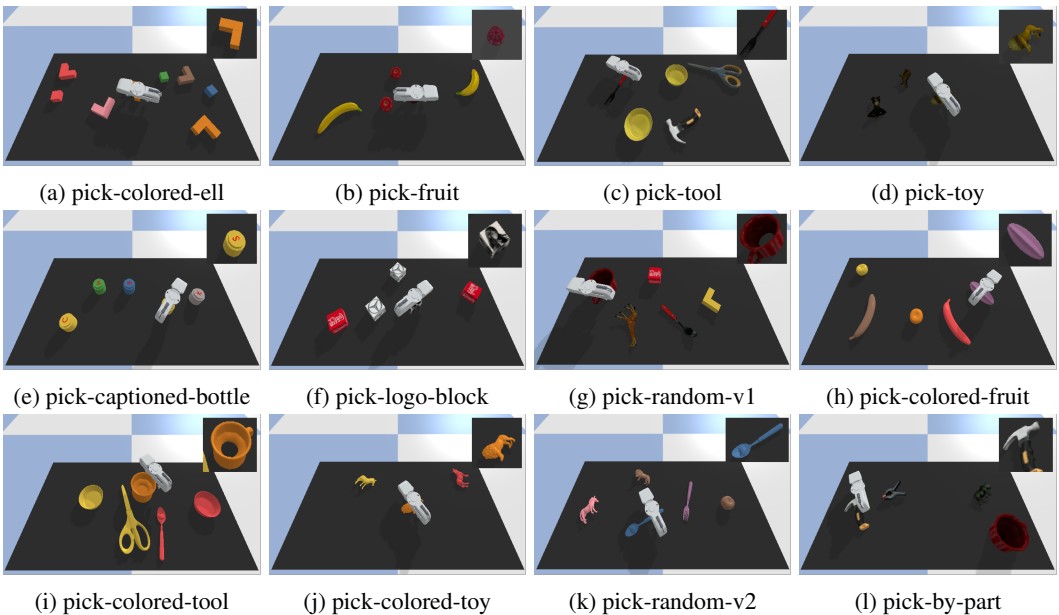

| (a) pick-colored-ell | (b) pick-fruit | (c) pick-tool | (d) pick-toy |
| (e) pick-captioned-bottle | (f) pick-logo-block | (g) pick-random-v1 | (h) pick-colored-fruit |
| (i) pick-colored-tool | (j) pick-colored-toy | (k) pick-random-v2 | (l) pick-by-part |

Figure 3: **Language-Grasp Benchmark.** Each subfigure describes a randomly initialized environment of each task. The goal object that the robot is currently instructed to grasp is highlighted at the top-right corner.

cameras pointing toward a 1m × 0.5m workspace. The corresponding language instruction to each expert action has 20 variations generated by ChatGPT-3.5, e.g., "pick up a mug by the handle", "locate and take a mug with the handle". A detailed description of the language instruction can be found in Appendix A.6.

### 4.1.1 TASK DESCRIPTION

As shown in Figure 3, *Language-Grasp Benchmark* has 10 fundamental language-conditioned tasks with 3 variations to evaluate the performance of language-conditioned grasp with a different focus. To get a reward, the agent must successfully grasp the object and satisfy the language instruction. The tasks could be split into two categories. The V-1 tasks allow only one variable, e.g., the color difference, shape difference, or pattern difference. The V-2 tasks allow two variables, e.g., the color difference combined with the shape difference, or the shape difference combined with the part difference. Here, we provide a short description of each task and a detailed description can be found in Appendix A.6.

**V-1 Tasks.** 1). *pick-colored-ell:* Pick the L-shape block with the instructed color. 2,3,4). *pick-fruit, pick-tool, pick-toy:* For each of the three tasks, objects from the same category are randomly placed on the workspace. The goal is to successfully grasp the object instructed by the language instruction. 5). *pick-caption:* Pick the bottle with the right caption on its cap. 6). *pick-logo-block.* The goal is to pick a block painted with the correct logo, e.g., "fetch a block with fire logo". 7). *pick-random-v1:* Five objects are randomly sampled from task 1 to task 6 and the goal is to pick the right object following the language.

**V-2 Tasks.** 8,9,10). *pick-colored-fruit,pick-colored-tool,pick-colored-toy:* Objects from the same category are colored differently and the goal is to grasp the correct object with the correct color. 11). *pick-random-v2:* Randomly sample seen combinations of colors and objects from tasks 8, 9, and 10. 12). *pick-novel-v2:* Randomly sample unseen combinations but seen colors and objects from tasks 8,9, and 10. There are 220 combinations across 22 different shapes and 10 different colors in tasks 8,9, and 10. A number of 45 combinations are novel. 13.) *pick-by-part:* The goal is to pick the instructed object by the instructed part, e.g., "pick up a mug by the handle".

**Settings and Metrics.** Every episode of each task is initialized with randomly placed n objects ($n \in \{4, 5\}$ due to the size of the workspace). A partial reward is assigned for each successful

| Model | pick-colored-ell | | pick-fruit | | pick-tool | | pick-toy | | pick-caption | | pick-logo | | pick-random-v1 | |
|---|---|---|---|---|---|---|---|---|---|---|---|---|---|---|
| | 1 | 10 | 1 | 10 | 1 | 10 | 1 | 10 | 1 | 10 | 1 | 10 | 1 | 10 |
| LEG-UNet (ours) | **65.0** | 76.5 | **76.5** | 88.5 | **91.0** | **94.0** | **98.2** | **99.5** | 59.7 | **60.7** | 66.2 | 80.7 | **88.2** | **94.8** |
| FCGQ | 38.5 | 64.7 | 66.2 | 79.5 | 74.7 | 88.5 | 67.5 | 87.7 | 60.2 | 59.7 | 51.0 | 54.7 | 74.2 | 86.8 |
| ViT | 14.2 | 8.75 | 67.7 | 73.75 | 41.5 | 61.75 | 41.0 | 39.5 | 55.0 | 59.75 | 47.2 | 50.0 | 40.0 | 52.6 |
| LEG-Cliport (ours) | 54.7 | **79.5** | 74.2 | **88.7** | 86.7 | 93.7 | 92.7 | 96.7 | **62.5** | 60.0 | **68.2** | 82.7 | 82.4 | 93.8 |
| Cliport-RN50 | 54.7 | 72.2 | 71.0 | 81.2 | 79.5 | 89.2 | 70.75 | 73.5 | 58.7 | 57.2 | 63.5 | **89.0** | 73.4 | 91.8 |
| Cliport-ViT | 40.7 | 57.5 | 63.2 | 62.5 | 58.5 | 56.5 | 50.5 | 57.0 | 57.7 | 59.5 | 53.0 | 57.5 | 56.6 | 44.8 |

Table 1: **Performance comparisons on Language-Grasp Benchmark V1 tasks (%)** on 100 tests v.s. the number of demonstration episodes (1, 10) used in training. Best performances are highlighted in bold.

| Model | pick-colored-fruit | | pick-colored-tool | | pick-colored-toy | | pick-random-v2 | | pick-novel-v2 | | pick-by-part[★] | |
|---|---|---|---|---|---|---|---|---|---|---|---|---|
| | 1 | 10 | 1 | 10 | 1 | 10 | 1 | 10 | 1 | 10 | 1 | 10 |
| LEG-UNet (ours) | 69.7 | 83.2 | 59.2 | 79.5 | 70.7 | 92.2 | 60.0 | 87.2 | 50.6 | 74.2 | **87.0** | **98.0** |
| FCGQ | 61.2 | 79.0 | 59.0 | 78.5 | 52.2 | 81.5 | 50.4 | 84.4 | 50.2 | 69.6 | 66.0 | 95.0 |
| ViT | 57.7 | 70.7 | 33.0 | 54.5 | 39.0 | 64.7 | 31.2 | 61.8 | 22.0 | 42.4 | 19.5 | 53.0 |
| LEG-Cliport (ours) | **72.5** | **86.5** | **65.0** | **89.0** | **72.2** | **94.0** | **67.4** | **92.2** | **60.0** | **85.0** | 81.5 | 97.7 |
| Cliport-RN50 | 58.0 | 76.5 | 45.0 | 80.2 | 38.2 | 75.5 | 41.0 | 82.6 | 42.6 | 74.6 | 50.0 | 94.7 |
| Cliport-ViT | 50.0 | 51.2 | 40.0 | 53.0 | 30.5 | 44.7 | 32.2 | 44.6 | 34.8 | 47.2 | 30.5 | 30.7 |

Table 2: **Performance comparisons on Language-Grasp Benchmark V2 tasks (%)** on 100 tests v.s. the number of demonstration episodes (1, 10) used in training. Best performances are highlighted in bold.

grasp. For example, if there are 4 toys presented in the workspace, each successful grasp will be credited a reward of 0.25. The successful grasp is defined as the grasp lifting the object and satisfying the language goal. A maximum of $n + 1$ grasping trials is set for each task.

**Demonstration.** Each demonstration of the task contains a set of observation-language-action triples $(o_t, \ell_t, \bar{a}_t)$, where $\bar{a}_t$ denotes the expert pick action. For V-1 tasks, one demonstration of each task covers all the objects exactly once, while for V-2 tasks, one demonstration covers different objects and different colors instead of the entire combinations roughly once. For instance, one demonstration of *pick-random-v1* contains 47 grasps that iterate different colored ell-shape blocks, different fruits, etc. One demonstration of *pick-random-v2* includes around 22 grasps that cover the 22 different shapes and 10 different colors. For *pick-by-part*, one demo contains 23 grasps that include the grasp of each part of each object once.

## 4.2 IMPLEMENTATION AND BASELINE

We implement two variations of our proposed method. One is trained from scratch with the U-net as the visual-language encoder and we denote it as LEG-*UNet*. The other uses the vision-language encoder from Cliport (Shridhar et al., 2022) and we call it LEG-*Cliport*. We compare our models against several strong baselines. **Baselines trained from scratch:** *FCGQ:* It is a modified architecture of Satish et al. (2019) where the language embedding is added into the bottleneck of the FCN. This is an FCN with a 36-channel output that associates each grasp rotation to a channel of the output. *ViT:* Inspired by Wang et al. (2022e); Shridhar et al. (2023), we use a Transformer to encode image grids and language tokens and then reshape the tokens spatially and use FCN and upsampling layers to generate the n-channel dense feature map. **Baselines with pretrained models:** *Cliport-RN50:* It take use the pick module of Cliport (Shridhar et al., 2022) and outputs the n-channel dense feature map. *Cliport-ViT:* It replaces the visual encoder of *Cliport-RN50* with pretrained Clip-ViT32 visual encoder.

## 4.3 TRAINING AND TESTING DETAILS

We assume access to a dataset $\mathcal{D} = \{\zeta_1, \zeta_2, ..., \zeta_n\}$ of n expert demonstrations, where each demo $\zeta_i = \{(o_1, \ell_1, \bar{a}_1), (o_2, \ell_2, \bar{a}_2), ..., (o_m, \ell_m, \bar{a}_m)\}$ is a set of successful language grasps that each one of $m$ objects in the environment is grasped once. The model of each method is trained with a dataset of $\{1, 10\}$ expert demonstrations on *pick-random-v1* (47 grasps each demo), *pick-random-v2*

(around 22 grasps each demo) and *pick-part* (23 grasps each demo) individually. The model trained on *pick-v1-random* is tested on 100 different scenes of the V-1 tasks. Please note it is slightly different from a multi-task agent which has seen the distribution of each task during training. Similarly, models trained on *pick-random-v2* are tested on V-2 tasks with 100 test scenarios. The model trained on *pick-part* is measured on 100 test cases on the same task. The orientation resolution is set as $\frac{\pi}{36}$ for all methods. We report the training time and GPU memory requirement for each method in Appendix A.1. We use the expert action to generate one hot map as the ground truth label. Cross-entropy loss is used to train the model end to end. Each model is trained with the number of 40k SGD steps and we test the model every 10k steps. We report the best performance for each model.

## 4.4 RESULTS ON SIMULATED TASKS

Table 1 shows the results of all the methods trained with $\{1, 10\}$ demos of the pick-random-v1 task and tested on 100 unseen configurations on all the V-1 tasks. Table 2 shows all the methods trained with $\{1, 10\}$ demos of the pick-random-v2 task and tested on 100 unseen configurations on all the V-2 tasks. Please note that we train and test a single-task policy on the pick-by-part task. Several conclusions could draw from Table 1 and 2.

**LEG v.s. Others**. As shown in Table 1 and Table 2, with 1 demo available, LEG outperforms all baselines on the 13 tasks. It can also achieve above $90\%$ success rate on *pick-tool* and *pick-toy*. With 10 demos available, LEG outperforms all baselines on the 12 tasks. It hits above $90\%$ success rate on 6 tasks. It indicates the sample efficiency as well as the compelling success rate of our proposed method.

**Pretrained-LEG v.s. Training From Scratch**. For tasks with shape difference only, the train-from-scratch LEG-UNet is overall better than our per-trained LEG-Cliport. For tasks involved with colors, captions, and logos, our pretrained model shows better performance and generalization ability, especially on *pick-novel-v2* that requires the model to distinguish novel geometric-semantic combinations.

**ViT Backbone v.s. FCN Backbone**. The Fully Convolutional Network (FCN) backbone variations are overall better than the ViT backbone variations by a large margin. Since ViT processes the image by flattening the patches and utilizing self-attention layers, it breaks the well-formatted pixel structure and also loses the translational equivariance compared with FCNs. The observation aligns with that geometric features and symmetries are more vital in learning robot skills.

## 4.5 REAL-WORLD EXPERIMENTS

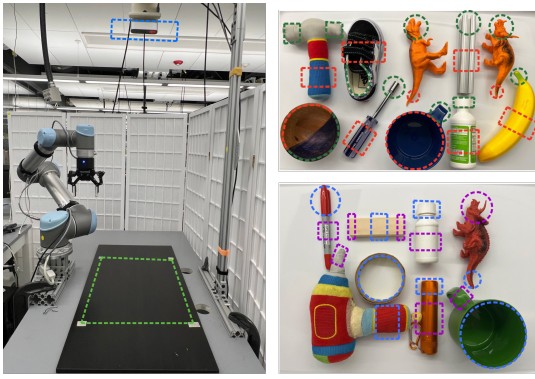

We evaluated LEG-*UNet* on *pick-by-part* with a physical robot - all demonstrations were performed on the real robot. As shown in Figure 4, we used a UR5 robot with a Robotiq-85 end effector. The workspace was defined as a 35 cm $\times$ 48 cm region on a table and the observation $o_t$ were $208 \times 288$ RGB-D images captured by an Azure Kinect sensor that was mounted pointing directly down to the table. We obtained 5 human demonstrations with a total of 100 grasps of *pick-by-part* on a set of 10 objects. Each object was labeled with two parts, as shown in the top-right image of Figure 4 and each demonstration contains 20 grasps, i.e., one grasp per part per object.

Figure 4: **Real World Experiment.** The left figure shows the robot setting and the right figures show seen (top) and novel objects (bottom)

**Training and Testing.** We trained the LEG-*UNet* with $\{1, 5\}$ demonstrations for 40k SGD steps. During testing 5 objects were randomly selected and placed on the table without replacement. We evaluate the model with 8 runs (a total of 40 grasps and 2 grasps per part). We report three metrics: the success rate of grasping, the success rate of grasping the instructed object, and the success rate of grasping the instructed object by the instructed part. We also tested our model trained with 5 demos

|  | grasp success rate | grasp correct object | grasp correct object-part |
|---|---|---|---|
| 8 unseen objects | 83.3% (25/30) | 56.6% (17/30) | 46.6% (14/30) |
| Pick-part-1 demo | 82.5% (33/40) | 75.0% (30/40) | 62.5% (25/40) |
| Pick-part-5 demos | 97.5% (39/40) | 92.5% (37/40) | 90.0% (36/40) |

Table 3: **Real World Experiments**

on 8 novel objects, as shown in the bottom image of Figure 4. A detailed description of the objects and demonstration can be found in Appendix A.5.1.

**Results on Real-Robot Experiments.** Table 3 shows the results of the *pick-by-part* experiments. With only 5 demos, our method can achieve a 97.5% grasping success rate and 90.0% success rate of grasping the correct object by the correct part. The results of 8 runs of the model trained with 1 demo are lower than those of simulated experiments. This is likely caused by the fact that 1). Real sensor noises and lighting affected the results 2). A single top-down camera cannot provide a perfect orthographic projection of the workspace. As shown in the first row of Table 4, the model trained with 5 demos can generalize to objects with novel shapes and names. Similar objects own similar language embeddings and generate similar functional dynamic kernels. Videos of the real-robot experiment are included in the supplementary materials.

## 4.6 ABLATION STUDY

|  | pick-by-part | pick-random-V1 | pick-random-V2 | pick-novel-V2 |
|---|---|---|---|---|
| LEG-UNet | **98.0** | **94.8** | 87.2 | 74.2 |
| kernel w.o Lan Condi. | 92.6 ($\downarrow$ 6.8) | 90.0($\downarrow$ 4.8) | 85.0 ($\downarrow$ 2.2) | 75.8 ($\uparrow$ 1.6) |
| kernel w.o Steerablility | 90.7 ($\downarrow$ 7.3) | 85.8 ($\downarrow$ 9.0) | 80.2 ($\downarrow$ 7.0) | 62.0 ($\downarrow$ 12.2) |
| $\phi$ w.o Lan Emd | 97.0 ($\downarrow$ 1.0) | 92.6 ($\downarrow$ 1.8) | **88.4**($\uparrow$ 1.2) | **78.2**($\uparrow$ 4.0) |

Table 4: **Ablation Study.** Arrows indicate the performance difference between LEG-UNet and each three ablation variations.

To investigate the relative importance of the steerability and language-conditioned property of our dynamic kernel, we design 3 variations of LEG-*UNet*: 1) LEG-UNet without language condition. We remove the language embedding from the dynamic kernel generator. Instead, we randomly initialize the trainable parameters and feed it the $\psi$ network. It will generate an unconditioned steerable kernel. 2) LEG-UNet without steerability. We keep the language embedding for the $\phi$ network but remove the lifting and Fourier Transformation to generate a language-conditioned non-steerable kernel. 3). We remove the language input to the $\phi$ network and the top branch only takes the image as the input while the language-conditioned kernel remains. All the models are trained with 10 demos on the tasks of pick-by-part, pick-random-v1, and pick-random-v2 individually. We report the results in Table 4. We find that 1) Without the language-condition kernel generator, the performance overall drops, especially for the pick-by-part. It shows that the generated kernel with aligned language feature is better than the unconditioned kernel ; 2). Without the steerability, the success rate drops significantly due to the lack of the local SO(2) equivariance; 3). Without the language embedding to the $\phi$ branch, there are trade-offs related to tasks. The ablation study suggests the importance of the two key components in LEG-*UNet*, the language-conditioned property and the steerability property.

## 5 CONCLUSION

In this work, we analyze the symmetry of the language-condition grasping and propose Language-conditioned Equivariant Grasp (LEG). It realizes the underlying symmetry of the task with a flexible two-branch design and links the language instruction in the format of a steerable dynamic kernel to visual features. We further present a language-conditioned grasp benchmark and evaluate the performance of our method and baselines on various tasks. Our proposed method demonstrates a strong inductive bias on sampling efficiency and a high success rate in language-conditioned grasp. Finally, we demonstrate that the method can effectively learn manipulation policy on a physical robot and can generalize to novel objects. Distilling the large models to low-level manipulation skills still has a long way to go. Using the diffusion model to generate steerable kernels and extending our method to 3D language-conditioned grasping are interesting future directions.

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

# A APPENDIX

## A.1 TRAINING TIME AND GPU MEMORY AND NUMBER OF PARAMETERS.

We report the number of parameters of our methods and baselines in Table 5. Another possible baseline is *Cliport-Rotator* which is similar to *Cliport-UNet* but feeds a stack of n rotated images to the visual encoder and generates a one-channel dense feature per image. The results are then counter-rotated at the network's output and each channel corresponds to one pick orientation (Shridhar et al., 2022). However, it has a heavy computation load to process a batch of n images. Since the rotation resolution of our action space is $\frac{\pi}{36}$, it requires a stack of 72 differently rotated images to train *Cliport-Rotator*, which excess the limit of GPU memory we have access to. To compare it with others, we also report the number of GPU memory to train *Cliport-Rotator* with a batch of 36 rotated images.

| | LEG-*UNet* | *Uet* | *ViT* | LEG-*Cliport* | *Cliport-Unet* | *Cliport-ViT* | *Cliport-rotator-36*[*] |
|---|---|---|---|---|---|---|---|
| # of parameters (M) | 3.3 | 6.8 | 14 | 63 | 94 | 61 | 61 |
| GPU memory (GB) | 2.6 | 2.3 | 2.4 | 3.9 | 4.1 | 3.4 | 14 |
| training time/step (s) | 0.12 | 0.09 | 0.1 | 0.16 | 0.14 | 0.12 | 1.3 |

Table 5: **Memory and computation time.** Test on NVIDIA 3090.

## A.2 PROOF

### A.2.1 BACKGROUND ON SYMMETRY GROUPS

**Group and Representation.** $SO(2)$ contains the continuous planar rotations $\{\text{Rot}_\theta : 0 \leq \theta < 2\pi\}$. $C_n = \{\text{Rot}_\theta : \theta \in \{\frac{2\pi i}{n} | 0 \leq i < n\}\}$ contains only rotations by angles which are multiples of $2\pi/n$. A $d$-dimensional *representation* $\rho \colon G \to \text{GL}_d$ of a group $G$ assigns to each element $g \in G$ an invertible $d \times d$-matrix $\rho(g)$. Different representations of $SO(2)$ or $C_n$ help to describe how different signals are transformed under rotations.

1. The trivial representation $\rho_0 \colon SO(2) \to \text{GL}_1$ assigns $\rho_0(g) = 1$ for all $g \in G$, i.e. no transformation under rotation.

2. The standard representation

$$\rho_1(\text{Rot}_\theta) = \begin{pmatrix} \cos\theta & -\sin\theta \\ \sin\theta & \cos\theta \end{pmatrix}$$

   represents each group element by its standard rotation matrix. Notice that $\rho_0$ and $\rho_1$ can be used to represent elements from either $SO(2)$ or $C_n$.

3. The regular representation $\rho_{\text{reg}}$ of $C_n$ acts on a vector in $\mathbb{R}^n$ by cyclically permuting its coordinates $\rho_\lambda(\text{Rot}_{2\pi/n})(x_0, x_1, ..., x_{n-2}, x_{n-1}) = (x_{n-1}, x_0, x_1, ..., x_{n-2})$.

4. The irreducible representation $\rho_{\text{irrep}}^i$ could be considered as the basis function with the order/frequency of $i$, such that any representation $\rho$ of $G$ could be decomposed as a *direct sum* of them. Signals defined on the group $SO(2)$ can be decomposed as limits of linear combinations of complex exponential functions $(\sin, \cos)$.

**Feature Vector Field.** We formalize images and 2D feature maps as feature vector fields, i.e., functions $f \colon \mathbb{R}^2 \to \mathbb{R}^c$, which assign a feature vector $f(\mathbf{x}) \in \mathbb{R}^c$ to each position $\mathbf{x} \in \mathbb{R}^2$. The action of an element $g \in SO(2)$ on $f$ is a combination of a rotation in the domain of $f$ via $\rho_1$ (this rotates the pixel positions) and a transformation in the channel space $\mathbb{R}^c$ (i.e., fiber space) by $\rho \in \{\rho_0, \rho_1, \rho_\lambda, \rho_{\text{irrep}}\}$. If $\rho = \rho_0$, the channels do not change. If $\rho = \rho_{\text{reg}}$, then the channels cyclically permute according to the rotation. If $\rho = \rho_{\text{irrep}}$, then the channels shift.

We denote this action (the action of $g$ on $f$ via $\rho$) by $T_g^\rho(f)$:

$$[T_g^\rho(f)](\mathbf{x}) = \rho(g) \cdot f(\rho_1(g)^{-1}\mathbf{x}). \tag{5}$$

**Equivariant Mapping and Steerable Kernels** A function $F$ is equivariant if it commutes with the action of the group,

$$T_g^{\text{out}}[F(f)] = F(T_g^{\text{in}}[f]) \tag{6}$$

where $T_g^{\text{in}}$ transforms the input to $F$ by the group element $g$ while $T_g^{\text{out}}$ transforms the output of $F$ by $g$. The most equivariant mappings between spaces of feature fields are *convolutions with G-steerable kernels* (Weiler et al., 2018; Jenner & Weiler, 2021). Denote the input field type as $\rho_{\text{in}} \colon G \to \mathbb{R}^{d_{\text{in}} \times d_{\text{in}}}$ and the output field type as $\rho_{\text{out}} \colon G \to \mathbb{R}^{d_{\text{out}} \times d_{\text{out}}}$. The G-steerable kernels are convolution kernels $K \colon \mathbb{R}^n \to \mathbb{R}^{d_{\text{out}} \times d_{\text{in}}}$ satisfying the *steerability constraint*, where $n$ is the dimensionality of the space

$$K(g \cdot x) = \rho_{\text{out}}(g) K(x) \rho_{\text{in}}(g)^{-1} \tag{7}$$

**SO(2) Fourier Transformation.** Signals defined over the group $\text{SO}(2)$ can be decomposed as limits of linear combinations of complex exponential functions (for $\text{SO}(2)$). We refer to the Fourier transform that maps $\text{SO}(2)$-signals to the coefficients of the basis functions as $\mathcal{F}^+$ and the inverse Fourier Transform as $\mathcal{F}^{-1}$.

### A.2.2 PROOF OF PROPOSITION 1

**Translational Equivariance.** Since FCNs are translationally equivariant by their nature, if the target object $b^\ell$ is translated to a new location, the cross-correlation between $\kappa(\ell_t) * \phi(o_t, \ell_t)$ will capture this translation and there is no change in the change space.

**Rotation Equivariance.** Assuming $\phi$ satisfies the equivariant property that $\phi(T_g^0 o_t, \ell_t) = T_g^0 \phi(o_t, \ell_t)$ and the rotation of $b^\ell$ is represented by $T_g^0 o_t$, we start the proof with lemma 1 and lemma 2.

**Lemma 1** *if $k(x)$ is a steerable kernel that takes trivial-type input signal, it satisfies $T_g^0 K(x) = \rho_{\text{out}}(g^{-1}) K(x)$.*

**Prove Lemma 1.** $\rho_0(g)$ is an identity mapping. Substituting $\rho_{\text{in}}$ with $\rho_0(g)$ and $g^{-1}$ with $g$ in Equation 7

$$
\begin{aligned}
T_g^0 K(x) &= K(g^{-1}x) \\
&= \rho_{\text{out}}(g^{-1}) K(x) \rho_{\text{in}}(g) \\
&= \rho_{\text{out}}(g^{-1}) K(x)
\end{aligned}
$$

**Lemma 2** *Cross-correlation satisfies that*

$$(T_g^0(K \star f))(\vec{v}) = ((T_g^0 K) \star (T_g^0 f))(\vec{v}) \tag{8}$$

**Prove Lemma 2.** We evaluate the left-hand side of Equation:

$$T_g^0(K \star f)(\vec{v}) = \sum_{\vec{w} \in \mathbb{Z}^2} f(g^{-1}\vec{v} + \vec{w}) K(\vec{w}).$$

Re-indexing the sum with $\vec{y} = g\vec{w}$,

$$= \sum_{\vec{y} \in \mathbb{Z}^2} f(g^{-1}\vec{v} + g^{-1}\vec{y}) K(g^{-1}\vec{y})$$

is by definition

$$
\begin{aligned}
&= \sum_{\vec{y} \in \mathbb{Z}^2} (T_g^0 f)(\vec{v} + \vec{y})(T_g^0 K)(\vec{y}) \\
&= ((T_g^0 K) \star (T_g^0 f))(\vec{v})
\end{aligned}
$$

as desired.

Given Lemma 1 and lemma 2, we can prove that

$$
\begin{aligned}
\kappa(\ell_t) * \phi(T_g^0 o_t, \ell_t) =& \kappa(\ell_t) * T_g^0 \phi(o_t, \ell_t) \\
=& \kappa(\ell_t) * T_g^0 \phi(o_t, \ell_t) \\
=& T_g^0 T_{g^{-1}}^0 \kappa(\ell_t) * T_g^0 \phi(o_t, \ell_t) \\
=& T_g^0 [T_{g^{-1}}^0 \kappa(\ell_t) * \phi(o_t, \ell_t)] \text{ lemma 2} \\
=& T_g^0 [\rho_{\text{out}}(g)\kappa(\ell_t) * \phi(o_t, \ell_t)] \text{ lemma 1}
\end{aligned}
$$

It states that if there is a rotation on $o_t$, the grasp position is changed by $T_g^0$, and the rotation is changed by $\rho_{\text{out}}(g)$. Since the cross-correlation is calculated for each pixel without stride, the rotated $b^\ell$ is captured by $\rho(g)$. In our implementation, we generate the language-conditioned steerable kernel $\kappa(\ell_t)$ but remove the constraint of the equivariant property of $\phi$. However, the U-Net architecture with the long skip connection can maintain the equivariance a little bit, and extensive data augmentation is used to force the model to learn the equivariance.

### A.2.3 PROOF OF THE STEERABILITY OF $\mathcal{L}(\psi(\cdot))$

$$
\begin{aligned}
\mathcal{L}(T_g^0 \psi(\cdot)) =& T_g^0 \{T_{g_1}^0 \psi(\cdot), T_{g_2}^0 \psi(\cdot) \cdots, T_{g_n}^0 \psi(\cdot)\} \quad g_i \in C_n \\
=& \{T_{gg_1}^0 \psi(\cdot), T_{gg_2}^0 \psi(\cdot) \cdots, T_{gg_n}^0 \psi(\cdot)\} \\
=& \{T_{g_2}^0 \psi(\cdot), T_{g_3}^0 \psi(\cdot) \cdots, T_{g_n}^0 \psi(\cdot), T_{g_1}^0 \psi(\cdot)\} \text{ if } g = g_1 \\
=& \rho_{\text{reg}}(g^{-1})\mathcal{L}(\psi(\cdot))
\end{aligned}
$$

Since $\mathcal{L}(T_g^0 \psi(\cdot)) = \mathcal{L}(g^{-1}x)$, we achieve that $\mathcal{L}(g^{-1}x) = \rho_{\text{reg}}(g^{-1})\mathcal{L}(x)$. Substituting $g^{-1}$ with $g$ shows that $\kappa(c) = L(\psi(\cdot))$ satisfies the steerability constraint shown in Equation 7 and it is a steerable kernel with regular-type output and trivial-type input. Since Fourier transformation on the channel space maps the discrete $SO(2)$ signal above each pixel to the coefficients of the basis function. It realizes an irreducible steerable kernel that has trivial-type input and irrep-type output (Weiler & Cesa, 2019; Cesa et al., 2021).

### A.3 NETWORKS.

$\phi$ **Network.** The green branch in Figure 2 is the $\phi$ network which encodes the current observation and the language instruction jointly. $\phi$ network is a U-Net architecture with concatenated language embeddings from CLIP (Radford et al., 2021) in its intermediate features. Since the $\phi$ network is a function that takes images and instructions and outputs feature maps, it is flexible to be replaced with any modern pretrain VLM to leverage the prior knowledge from large datasets.

**Language-conditioned dynamic Kernel Generator $\kappa$.** As shown in Figure 2, the bottom yellow branch illustrate our language kernel generator. There are two main phases: generation and lifting. During generation, we repeat the pretrained instruction embedding from CLIP text encoder (Radford et al., 2021) to form a 2D tensor with spatial dimensions. Then, the 2D feature is then fed into a U-Net to generate a 3-channel trivial feature map $\psi(\ell_t)$. After that, we lift the $\psi(\ell_t)$ with $C_{90}$, i.e., we rotate the $\psi(\ell_t)$ 90 times and generate a stack of rotated features. As a result, there is 90-dimension channel feature above each pixel. Finally, we apply Fourier transformation on the channel space and represent the $SO(2)$ with coefficients of the basis functions.

**Language as input for both $\phi$ and $\kappa$.** One thing we would like to address is that the $\phi$ and the $\kappa$ module both take language as input. It is more like a self-attention mechanism during the convolution between the steerable kernels and the attention logits rather than repetitive language inputs. We notice that the language input on both branches is overall beneficial and stable during training.

**Cliport-rotator Baseline.** In order to demonstrate the efficiency of the rotation equivariance compared with the pre-rotation technique which has shown effective for rotation generalization (Zeng et al., 2021), we implement a baseline called Cliport-Rotator. The idea is to rotate our observation

36 times for every ten degrees and to rotate then back at the end so that the argmax rotation angle can be obtained. However, in order to store all the padded and rotated observations, it is very memory-heavy, which makes it impractical for further investigation.

## A.4 Real-world Experiment details.

**Demo Collection:** In the real-world demo collection, we collect a total of five demonstrations. Every single demonstration is defined as 20 grasps, i.e., one grasp per part of each of the 10 training objects as shown in the top-right photo in Figure 4.

The labels (actions) are created from scratch by controlling the UR5 robot arm manually and placing it in the correct orientation and position for a successful grasping of the goal object. Meanwhile, an instruction of the goal object is also given. Worth mentioning, that our real-world model is only trained on real-world data instead of doing sim-to-real transfer.

**Testing the Model:** After the training process is completed, we test performance by giving the model an observation and an instruction on randomly initialized objects. During the testing process, we did not move the robot arm manually but purely input the instructions via the terminal such as "Grasp the middle part of the cup." and the model outputs the action then completes the grasping with a predefined motion planner.

**Failure Mode of Seen Objects:** Although performance in training sets is good, there are two grasping failures:

- **Wrong part:** The goal is to grasp the cup rim → grasp the cup's handle.
- **Wrong part:** The goal is to grasp the shoe heel → grasp the shoe's middle part

**Failure Mode of Novel Objects:** The error rate on unseen objects increases with the unseen instructions and unseen geometric shapes. Part of the failure examples including the instructions and the consequences are demonstrated below:

- **Grasping failure:** aim for the middle of the pen → grasp a random pose on table
- **Wrong object:** aim for dinosaur's tail → obtained the pen
- **Wrong object:** aim for the green mug's handle → get the flashlight
- **Wrong part:** aim for the green mug's handle → locate on its rim section

**Failure Analysis:** Obviously, the failure rate is affected by the objects' geometric pattern. Specifically, for example, both the shoe's heel and the cup's rim have a "ring"-like pattern so the model fails to distinguish these two which indicates that the model focuses more on geometric patterns instead of semantic patterns in this case.

Besides the object's features, interrelationships between objects in different groups also affect the grasping success rate. For instance, the successful grasping of the green mug happens after we move the flashlight according to the fact that the flashlight's material is more reflective than the mug's, producing more unseen illumination patterns, and making the model fail to recognize the green mug. Statistically, the outcome tended to be influenced by the combination of objects in the same group via the aspect of brightness, color, and identical geometry.

## A.5 Object settings

### A.5.1 Object settings in simulation

We get 3D meshes from two sources, YCB dataset (Calli et al., 2015) and GraspNet dataset (Fang et al., 2020). Some of the large meshes, e.g. lion, elephant, etc., from GraspNet are simplified using Meshlab (Cignoni et al., 2008).

**Colors variations.** In some tasks, such as *pick-colored-ell* and *pick-colored-tool*, we set objects to have different colors. The colors are randomly selected from a pre-defined color set blue, red, green, orange, yellow, purple, pink, cyan, brown, gray.

| task name | Object set |
|---|---|
| (1) *pick-colored-ell* | ell shape block with color variations |
| (2) *pick-fruit* | banana, apple, strawberry, lemon, starfruit, avocado |
| (3) *pick-tool* | fork, knife, medium-clamp, mini-claw-hammer, mug, scissors, spoon, bowl |
| (4) *pick-toy* | deer, elephant, giraffe, hippo, lion, monkey, rhino, zebra |
| (5) *pick-caption* | bottles captioned by 'A', 'B', 'C', 'D', 'E', 'H', 'K', 'S' |
| (6) *pick-logo* | blocks textured by coke, benz, fire, husky, manga-girl, manga-boy |
| (7) *pick-random-v1* | objects from task (1)-(6) |
| (8) *pick-colored-fruit* | objects from *pick-fruit* with color variations |
| (9) *pick-colored-tool* | objects from *pick-tool* with color variations |
| (10) *pick-colored-toy* | objects from *pick-toy* with color variations |
| (11) *pick-random-v2* | objects from task (8-10) |
| (12) *pick-novel-v2* | objects from task (8-10) with novel color shape combinations |
| (13) *pick-by-part* | objects with part specifications: fork (head, middle, end), knife (head, middle, end), spoon (head, middle, end), medium-clamp (head, handle), mini-claw-hammer (head,handle), mug(brim,handle), scissors(handle,middle), lion (head,body), hippo (head, body), rhino (head, body) |

Table 6: Object sets in simulation tasks

### A.5.2 OBJECT SETTINGS IN REAL WORLD

All the data in real-world grasping experiments are collected from scratch by ourselves and are separated into two parts: normal training data and novel data. The training data contains 10 different items each with two parts such as *hammer's head, cup's handle, triceratops' tail, shoe's heel*, etc. Parallel to training data, the novel object settings contain 8 objects in training data: *mug/cup's handle, dinosaur's head, pen's middle part*, etc., and most parts of them share similar features as those in original training setting. We name it *novel objects* not only for the unseen features in geometry aspects but also in language's description such as *pen, mug, flashlight*, etc. which never appear in the training set. The specific descriptions for objects' names and their respective instruction examples can be found in Table 7.

### A.6 LANGUAGE INSTRUCTION PREPROCESSING

For our model, the performance depends on the assumption that the text embeddings should be similar for similar instructions, e.g. "*pick up the mug by its handle*" and "*grab the cup with its handle*". The text encoder of CLIP has the ability to distinguish language meanings on a certain level because it is pretrained on both text and image data. However, since no robot data is involved during pretraining, the text encoder is not necessarily able to produce similar embedding for the two aforementioned instructions. In order to close the gap between the pretraining data and the robot data in terms of language, we perform data augmentation on language instructions in our training set in order to ensure our steerable kernels are the same given similar instructions. The approach is to generate rephrased instruction templates based on a basic template, e.g. "*pick up the [object name] by the [part name]*". Using chatGPT (OpenAI, 2023), we generate synonym candidates and filter them manually. For each task, we provide 20 instruction templates as shown in Table 8

| Object name | Object part | Instruction example |
|---|---|---|
| **Training objects** | | |
| (1)Cup | Handle
Rim | Pick up a cup by the handle.
Retrieve a cup by the rim. |
| (2)Hammer | Head
Handle | Gather a hammer by the head.
Select a hammer by the handle. |
| (3)Triceratops | Head
Tail | Choose a Triceratops by the head.
Find and collect a Triceratops by the tail. |
| (4)Rex | Head
Tail | Locate and take a Rex by the head.
Collect a Rex by the tail. |
| (5)Banana | Root
Middle | Pick out a banana by the root from the bunch.
Procure a banana by the middle. |
| (6)Silver Bar | Side
Middle | Collect a silver bar by the middle.
Secure a silver bar by the middle. |
| (7)Shoe | Heel
Middle | Obtain a shoe by the heel.
Pluck a shoe by the middle. |
| (8)Screwdriver | Head
Handle | Gather a screw driver by the head.
Pick up a screw driver by the handle. |
| (9)Bottle | Body
Head | Find and collect a bottle by the body.
Retrieve a bottle by the head. |
| (10)Bowl | Brown part
Black part | Handpick a bowl by the brown part.
Acquire a bowl by the black part. |
| **Novel objects** | | |
| (1)Cup/Mug | Handle
Rim | Collect the cup/mug by the handle.
Obtain the cup/mug by the rim. |
| (2)Tape | Rim | Secure a tape by the rim. |
| (3)Dinosaur | Head
Tail | Pick out a dinosaur by the head from the bunch.
Acquire a dinosaur by the tail. |
| (4)Flashlight | Middle
Side | Harvest a flashlight by the middle.
Pluck a flashlight by the side. |
| (5)Hammer | Head
Handle | Find and collect a hammer by the head.
Collect a hammer by the handle . |
| (6)Bottle | Head
Body | Gather a bottle by the head.
Handpick a bottle by the body. |
| (7)Pen | Side
Middle | Retrieve a pen by the side.
Obtain a pen by the middle. |
| (8)Block | Side
Middle | Fetch a block by the side.
Secure a block by the middle. |

Table 7: Training and novel object sets in real robotic tasks

| Task | Language instruction templates |
|---|---|
| (1) *pick-color-ell* | *Pick up [color] [object name].* |
| (8) *pick-colored-fruit* | *Retrieve a [object name] of the [color] variety.* |
| (9) *pick-colored-tool* | *Gather a [color] [object name].* |
| (10) *pick-colored-tool* | *Select a [object name] that is [color].* |
| (11) *pick-random-v2* | *Choose [color] [object name] from the collection.* |
| | *Find and collect a [color] [object name].* |
| | *Acquire a [object name] in the shade of [color].* |
| | *Pluck a [object name] that is [color].* |
| | *Handpick a [color] [object name].* |
| | *Secure a [object name] with the color [color].* |
| | *Locate and take a [color] [object name].* |
| | *Fetch a [object name] in the [color] variant.* |
| | *Hand-select a [color] [object name].* |
| | *Procure a [object name] with the color [color].* |
| | *Harvest a [color] [object name].* |
| | *Obtain a [color] variety of [object name].* |
| | *Retrieve a [color] [object name] from the orchard.* |
| | *Pick out a [color] [object name] from the bunch.* |
| | *Collect a [object name] in [color] form.* |
| | *Search for and bring a [color] [object name].* |
| (13) *pick-by-part* | *Pick up a [object name] by the [part].* |
| | *Retrieve a [object name] by the [part].* |
| | *Gather a [object name] by the [part].* |
| | *Select a [object name] by the [part].* |
| | *Choose a [object name] by the [part] from the collection.* |
| | *Find and collect a [object name] by the [part].* |
| | *Acquire a [object name] by the [part].* |
| | *Pluck a [object name] by the [part].* |
| | *Handpick a [object name] by the [part].* |
| | *Secure a [object name] by the [part].* |
| | *Locate and take a [object name] by the [part].* |
| | *Fetch a [object name] in the by the [part].* |
| | *Hand-select a [object name] by the [part].* |
| | *Procure a [object name] by the [part].* |
| | *Harvest a [object name] by the [part].* |
| | *Obtain a [object name] by the [part].* |
| | *Retrieve a [object name] by the [part] from the orchard.* |
| | *Pick out a [object name] by the [part] from the bunch.* |
| | *Collect a [object name] by the [part].* |
| | *Search for and bring a [object name] by the [part].* |

Table 8: Language instruction templates of example tasks.

