# OpenReview forum: "Language Conditioned Equivariant Grasp"
_ICLR.cc/2024/Conference — Submitted to ICLR 2024_

### Official Review · Reviewer_fe34 · 2023-10-27

**Soundness:** 2 fair
**Presentation:** 2 fair
**Contribution:** 2 fair
**Rating:** 5
**Confidence:** 2

**Summary:**

This paper studies the challenge of learning a robotic policy to grasp objects based on natural language instructions. The authors introduced a Language-conditioned Equivariant Grasp (LEG) method that leverages CLIP features to align image and text observations and SO(2)-steerable kernels to improve sample efficiency. The effectiveness of LEG is tested on the Language-Grasp Benchmark, which consists of 10 varied language-driven grasping tasks. The method's efficiency and performance are also evaluated on an actual robot.

**Strengths:**

- The paper leverages a general frame to leverage the symmetry of language-conditioned grasping.
- The paper presented a dynamic kernel generator that maps language instructions to steerable kernels with rotation symmetry.
- The paper proposed a new grasping benchmark with ten categories of language conditions and corresponding expert demonstrations.
- The proposed methods show strong grasping performances on the proposed benchmark.

**Weaknesses:**

- The paper claims that the proposed inductive bias leads to high sample efficiency, but there is no direct evidence in the experiments.
- Neural Descriptor Field [1] is an SE(3) equivariant grasping method with a similar design to the proposed method. Please cite and analyze the similarities and differences between this paper and [1].
- A large body of recent papers (to name a few, [2-4]) in language-conditioned grasping are not included for comparison and analysis.
- The writing of this paper is unclear and uncareful.
    - The experiment section mentioned designing and obtaining a reward, suggesting that the proposed method is RL-based. However, there is no mention of reinforcement learning or any related formulation in the method section, except naming $p(a_t|o_t,l_t)$ as a policy once.
    - The paper did not explain how the model is trained. If it is an RL-based method, what are the observations and rewards, the algorithm used, and what hyperparameters are used? If not, what is the loss function, and how is it trained?
    - This paper has multiple typos; for example, at the end of page 3, there is a `\leq`, which I believe should be `\leg`.

[1] Simeonov, Anthony, et al. "Neural descriptor fields: Se (3)-equivariant object representations for manipulation." *2022 International Conference on Robotics and Automation (ICRA)*. IEEE, 2022.

[2] Xu, Yinzhen, et al. "Unidexgrasp: Universal robotic dexterous grasping via learning diverse proposal generation and goal-conditioned policy." *Proceedings of the IEEE/CVF Conference on Computer Vision and Pattern Recognition*. 2023.

[3] Sharma, Satvik, et al. "Language embedded radiance fields for zero-shot task-oriented grasping." *7th Annual Conference on Robot Learning*. 2023.

[4] Chen, Yiye, et al. "A joint network for grasp detection conditioned on natural language commands." *2021 IEEE International Conference on Robotics and Automation (ICRA)*. IEEE, 2021.

**Questions:**

- The introduction mentioned, “directly interleaving language features with image features breaks the geometric symmetries underlying the optimal policy”. Please elaborate on why interleaving features break symmetry.
- How are the gripper position and orientation obtained from the output feature map?
- The description of section 4.1 (For example, if there are 4 toys presented in the workspace, each successful grasp will be credited a reward of 0.25. The successful grasp is defined as the grasp lifting the object and satisfying the language goal. A maximum of n + 1 grasping trials is set for each task.) indicates that there are N objects in a scene, and the goal is to pick up each object individually, according to its language description. Only when all objects are picked up successfully can it obtain the full reward. Is my understanding correct?
- How is the grasping-by-part success rate measured? Is it by manual inspection or by some automated function?

---

> ### Author Response · Authors · 2023-11-21
>
> We express our gratitude to the reviewer for their valuable feedback on our manuscript. Here is our detailed response to the concerns raised. We believe that we have addressed all the major concerns that the reviewer had. We are happy to address more concerns given a score of 3.
>
> ## Weakness and Response
>
> ### Weakness 1:
>
> *The paper claims that the proposed inductive bias leads to high sample efficiency, but there is no direct evidence in the experiments.*
>
> ### Response 1:
>
> We respectfully disagree with the point raised. As shown in Tables 1 and 2, our proposed LEG-UNET and LEG-Cliport models outperform four strong baselines in 25 out of 26 testing scenarios, trained with either 1 or 10 demonstrations. Notably, our models trained with just a single demonstration can surpass the baselines trained with 10 demonstrations in the 'pick-tool' and 'pick-toy' tasks. Furthermore, they achieve a success rate above 90% in 6 out of 13 tasks. It is important to note that we trained our models exclusively on 'pick-random-v1' or 'pick-random-v2' and tested them on other V-1 or V-2 tasks.
>
> ### Weakness 2:
>
> *Neural Descriptor Field [1] is an SE(3) equivariant grasping method with a similar design to the proposed method. Please cite and analyze the similarities and differences between this paper.*
>
> ### Response 2:
>
> While NDF is commendable work, we believe that our design is not similar to it. Firstly, our focus is on language-conditioned grasping, learning directly from pixel space, whereas NDF does not incorporate language instructions and learns from point clouds. Secondly, our approach does not assume objectness, segmentation, or categorization, in contrast to NDF, which requires a segmentation module and learns a category-level pick-and-place policy. Thirdly, our method does not require any pretraining process, while NDF necessitates pretraining to acquire point descriptors. Although both methodologies learn from demonstrations and leverage task symmetries, it is intriguing why the reviewer perceives them as similar.
>
> ### Weakness 3:
>
> *A large body of recent papers (to name a few, [2-4]) in language-conditioned grasping are not included for comparison and analysis.*
>
> ### Response 3:
>
> In the revised related work section, we compare our proposed method with several recent works:
>
> 1. **Unidexgrasp** learns dexterous grasping from point cloud observations. Its approach to language-guided grasping involves projecting grasp proposals onto images and utilizing CLIP to compute the similarity between text and images. In contrast, our work is centered on language-conditioned grasping and proposes an end-to-end learning framework that effectively leverages task symmetries.
> 2. **Language-embedded Radiance Fields for Zero-shot Task-oriented Grasping** represents a very recent development. This method integrates language embeddings into the radiance field, highlighting regions according to language instructions. However, it uses GraspNet for grasp detection, and does not involve a learning process for grasping.
> 3. **A Joint Network for Grasp Detection Conditioned on Natural Language Commands** combines the visual vectors of sampled grasp regions with language vectors using element-wise products to regress the top-down grasp pose. Unlike our approach, this network is unable to generate a comprehensive distribution across the entire action space and cannot predict multiple optimal grasp candidates when they are available.
>
> These comparisons highlight the unique aspects and strengths of our approach in the context of the evolving landscape of language-conditioned robotic grasping. Here we provide a table (Table R3) to illustrate the key differences between these methods and ours.
>
> | TABLE R3. algorithm | language-conditioned | few-shot learning | extra-labeling-free | part grasping | end-to-end | Action space |
> | --- | --- | --- | --- | --- | --- | --- |
> | NDF[5] | ✘ | ✓ | ✘ | ✓ | ✓ | 6-DoF |
> | Unidexgrasp[2] | ✘ | ✘ | ✓ | ✘ | ✘ | 6-DoF |
> | LERF-TOGO [3] | ✓ | ✘ (zero-shot) | ✓ | ✓ | ✘ | 6-DoF |
> | CGNet[4] | ✓ | ✘ | ✓ | ✘ | ✓ | 4-DoF |
> | KITE [5] | ✓ | ✓ | ✘ (keypoint) | ✓ | ✘ | 6-DoF |
> | CROG [6] | ✓ | ✘ | ✘ (seg mask) | ✘ | ✓ | 4-DoF |
> | CLIPort [8] | ✓ | ✓ | ✓ | ✘ | ✓ | 4-DoF |
> | LEG (ours) | ✓ | ✓ | ✓ | ✓ | ✓ | 4-DoF |

---

> > ### Author Response · Authors · 2023-11-21
> >
> > ### Weakness 4:
> >
> > *The writing of this paper is unclear and uncareful.*
> >
> > ### Response 4:
> >
> > We would like to kindly point out that Reviewer FTbR and Reviewer n2g1 rated our presentation as good, and Reviewer n2g1 thinks "Writing is clear and easy to understand". Given the below clarification on the reward function and MDP assumption, we believe we have addressed the reviewer's concern about our presentation. If the review has more concerns, we are happy to further address them.
> >
> > ### Weakness 4.1:
> >
> > *The experiment section mentioned designing and obtaining a reward, suggesting that the proposed method is RL-based. However, there is no mention of reinforcement learning or any related formulation in the method section, except naming
> > as a policy once*
> >
> > ### Response 4.1:
> >
> > As mentioned in the first sentence of Section 3.1, our method employs behavior cloning or imitation learning. The use of reward and policy from the Markov Decision Process (MDP)[1] is a common practice in robotic learning and learning from expert demonstrations, and it does not imply that our method is based on Reinforcement Learning (RL). Additionally, as detailed in **Section 4.1.1, Settings and Metrics**, the reward function is utilized to assess the performance of the policy during the evaluation phases.
> >
> > ### Weakness 4.2:
> >
> > *The paper did not explain how the model is trained. If it is an RL-based method, what are the observations and rewards, the algorithm used, and what hyperparameters are used? If not, what is the loss function, and how is it trained?*
> >
> > ### Response 4.2:
> >
> > As stated in section 4.3 (training and testing details), We use the expert action to
> > generate one hot map as the ground truth label. Cross-entropy loss is used to train the model end
> > to end. Each model is trained with the number of 40k SGD steps and we test the model every 10k
> > steps. We report the best performance for each model.
> >
> > ### Weakness 4.3:
> >
> > *This paper has multiple typos.*
> >
> > ### Response 4.3:
> >
> > Thanks for pointing the typos out. We modified our paper accordingly.
> >
> > ## Questions and Response
> >
> > ### Questions 1:
> >
> > *The introduction mentioned, “directly interleaving language features with image features breaks the geometric symmetries underlying the optimal policy”. Please elaborate on why interleaving features break symmetry.*
> >
> > ### Response 1:
> >
> > The symmetry of the task is illustrated in Figure 1 and Equation 2. Directly combining the language feature with image features through concatenation, addition, or element-wise product cannot realize Equation 2.
> >
> > ### Questions 2:
> >
> > *How are the gripper position and orientation obtained from the output feature map?*
> >
> > ### Response 2:
> >
> > The output feature map is an n-channel dense feature map with the same spatial size as the top-down observation. As mentioned in section 3.1, since each pixel corresponds to a point in 3D space, the action $a_t$ is parameterized in terms of SE(2) coordinates $(u,v,\theta)$, where $$ $u$,$v$ denote the pixel coordinates of the gripper position and $\theta$ denotes the gripper orientation. The gripper orientation distribution is encoded as the N-channel feature above each pixel. Each channel corresponds to a $\frac{2\pi}{n}$ rotation angle.
> >
> > ### Questions 3:
> >
> > *The description of section 4.1 (For example, if there are 4 toys presented in the workspace, each successful grasp will be credited a reward of 0.25. The successful grasp is defined as the grasp lifting the object and satisfying the language goal. A maximum of n + 1 grasping trials is set for each task.) indicates that there are N objects in a scene, and the goal is to pick up each object individually, according to its language description. Only when all objects are picked up successfully can it obtain the full reward. Is my understanding correct?*
> >
> > ### Response 3:
> >
> > If there are 4 toys presented in the workspace, each successful grasp will be credited a reward of 0.25. Only when all objects are picked up successfully could it obtain the full reward. Otherwise, the total reward is calculated by the number of picked objects times 0.25.
> >
> > ### Questions 4:
> >
> > *How is the grasping-by-part success rate measured? Is it by manual inspection or by some automated function?*
> >
> > ### Response 4:
> >
> > In our simulated experiment, measurements are conducted automatically. In the real-world experiment, we manually measure them according to the annotations in Figure 4.

---

> > > ### Author Response · Authors · 2023-11-21
> > >
> > > ## Reference
> > >
> > > [1] BELLMAN, RICHARD. “A Markovian Decision Process.” Journal of Mathematics and Mechanics, vol. 6, no. 5, 1957, pp. 679–84. JSTOR, [http://www.jstor.org/stable/24900506](http://www.jstor.org/stable/24900506). Accessed 17 Nov. 2023.
> > >
> > > [2] Xu, Yinzhen, et al. "Unidexgrasp: Universal robotic dexterous grasping via learning diverse proposal generation and goal-conditioned policy." Proceedings of the IEEE/CVF Conference on Computer Vision and Pattern Recognition. 2023.
> > >
> > > [3] Sharma, Satvik, et al. "Language embedded radiance fields for zero-shot task-oriented grasping." 7th Annual Conference on Robot Learning. 2023.
> > >
> > > [4] Chen, Yiye, et al. "A joint network for grasp detection conditioned on natural language commands." 2021 IEEE International Conference on Robotics and Automation (ICRA). IEEE, 2021.
> > >
> > > [5] Simeonov, Anthony, et al. "Neural descriptor fields: Se (3)-equivariant object representations for manipulation." *2022 International Conference on Robotics and Automation (ICRA)*. IEEE, 2022.
> > >
> > > [6] Tziafas, Georgios, et al. "Language-guided Robot Grasping: CLIP-based Referring Grasp Synthesis in Clutter." arXiv preprint arXiv:2311.05779 (2023).
> > >
> > > [7] Satish, Vishal, Jeffrey Mahler, and Ken Goldberg. "On-policy dataset synthesis for learning robot grasping policies using fully convolutional deep networks." *IEEE Robotics and Automation Letters* 4.2 (2019): 1357-1364.
> > >
> > > [8] Shridhar, Mohit, Lucas Manuelli, and Dieter Fox. "Cliport: What and where pathways for robotic manipulation." Conference on Robot Learning. PMLR, 2022.

---

> ### Comment · Reviewer_fe34 · 2023-11-22
>
> Thank you for the response. The authors have addressed most of my concerns. However, I still believe that the writing of this paper makes it more difficult to follow than it should be. Based on that, I am raising my rating to 5 and lowering my confidence to 2. I am lowering my confidence because the main reason that I give a negative rating is for its writing. I'd be happy to raise my rating to 6 if this paper were reader-friendly.

---

> > ### Author Response · Authors · 2023-11-22
> >
> > The authors thank the reviewer for raising the score. According to the review's suggestions, we made two modifications (highlighted in blue) in our updated manuscript to further address review's concerns on the writing. (1) We added the related work in **Section 2** as the reviewer suggested. (2) We made two key modifications in **Section 1 and Section 4.3** to highlight that we are doing Learning from Demonstration (LfD), i.e., behavioral cloning, instead of reinforcement learning, as suggested.
> >
> > We are happy to address more concerns about writing if the review has further concerns or suggestions.

---

### Official Review · Reviewer_n2g1 · 2023-11-01

**Soundness:** 3 good
**Presentation:** 3 good
**Contribution:** 3 good
**Rating:** 6
**Confidence:** 2

**Summary:**

The paper proposes a new 2D equivariant grasp selection algorithm that's language conditioned. After discretizing output action to different buckets of angles, the paper propose to lifting 2D feature map to Fourier domain, in which conditioning embeddings like CLIP features are multiplied. The paper demonstrates the effectiveness of the method on simulated benchmarks as well as real robots. The paper also contributes a new benchmark on language conditioned grasps featuring 10 environments.

**Strengths:**

- The evaluation is solid by combining a variety of baselines, environments. It also features real-world grasping experiments featuring unknown objects.
- Previous method achieves equivariance at global observation level (rotating entire observation) while this paper focuses on local symmetry.
- Writing is clear and easy to understand

**Weaknesses:**

- The language part and equivariant part are not directly related to equivariance itself. I think what the paper tries to claim is finding a better way to condition and ground constrained networks here. However, starting with language conditioning can mislead the readerss to focus on the wrong thing.
- No equivariant baselines that simply takes in CLIP features by concatenation. If previous equivariant grasp models are not applicable here, please explain clearly why concatenation doesn't work.

**Questions:**

1. Do you have results for baselines in real-world experiments?
2. I am wondering how would the performance change according to the number of grasp bins.
3. In the writing, I hope the authors can state what's the input and output of the mapping that's equivariant to define the problem in a clearer way.

---

> ### Author Response · Authors · 2023-11-21
>
> We thank the reviewer for an especially detailed and thorough review. We provide a point-by-point response to the comments and questions below.
>
> ## Weakness and Response
>
> ### Weakness 1:
>
> *The language part and equivariant part are not directly related to equivariance itself. I think what the paper tries to claim is finding a better way to condition and ground constrained networks here. However, starting with language conditioning can mislead the readers to focus on the wrong thing.*
>
> ### Response 1:
>
> The primary focus of our work is to investigate efficient representations for grounding language in language-conditioned grasping tasks. As such, we place a strong emphasis on the language-conditioned steerable kernel, namely the language grounding aspect. Key points include:
>
> 1. **Language-Conditioned Policy**: Our policy is language-conditioned. This means that for a given image, the action distribution changes according to the language goal.
> 2. **SE(2) Equivariance and Dynamic Steerable Kernel**: The SE(2) equivariance, as stated in Equation 2, is achieved through a dynamic steerable kernel derived from the pre-trained language embedding. The process of generating this steerable kernel from the language embedding is a learnable aspect of our model.
> 3. **Steerable Kernel Generation**: To clarify, unlike previous approaches that may rely on E2CNN[1], our network does not constrain itself to achieve equivariance through such means. Instead, we generate the steerable kernel via lifting, a process detailed and validated in Appendix A.2.3.
>
> Our approach demonstrates a novel method of integrating language grounding with action planning in robotic tasks, leveraging the unique capabilities of dynamic steerable kernels.
>
> ### Weakness 2:
>
> *No equivariant baselines that simply takes in CLIP features by concatenation. If previous equivariant grasp models are not applicable here, please explain clearly why concatenation doesn't work.*
>
> ### Response 2:
>
> Excellent question. There are several key points to consider regarding the integration of language features with visual features:
>
> 1. **Local Symmetry**: Merely attaching the same language feature to each pixel is not sufficient to achieve local symmetry. This approach overlooks the complexity of how language features interact with spatially varying visual features.
> 2. **Inconsistency**: The direct concatenation of the language feature (trivial representation which is no change if there is a rotation) with equivariant visual feature (commonly, a regular representation which permutes if there is a rotation on the image) is not consistent.
> 3. **Data Augmentation**: Our baseline model, FCGQ, employs the UNet architecture. In this model, the hidden visual feature is concatenated with the language feature in the bottleneck layer. Moreover, the model's training involves extensive data augmentation, aligning with the variation suggested by the reviewer. The results in Table 1&2 show that our method with steerable kernels outperforms the concatenation baseline by a large margin.

---

> > ### Author Response · Authors · 2023-11-21
> >
> > ## Questions and Response
> >
> > ### Questions 1:
> >
> > Do you have results for baselines in real-world experiments?
> >
> > ### Responses 1:
> >
> > No. Given the comparison as shown in the pick-part task in Table 2, baselines are already underperforming with limited data (1/5 demonstrations) in simulation. Given the fact the performance will definitely drop with noisy observation in real world, we believe it is redundant to include baselines like CLIPort, FCGQ ,and ViT in real-world experiments. We chose the pick-part task for our real-world experiments because it represents a challenging yet intriguing aspect of robotic grasping. This task requires the model not only to pick the object but also to differentiate specific parts of it. And, we provide three metrics and experiments with different numbers of demonstrations to show the effectiveness of our method. We believe, given the difficulty of the task, various metrics, and the unseen-object evaluation, our physical experiments are sufficient to demonstrate the effectiveness of the real-world applicability of our method.
> >
> > ### Questions 2:
> >
> > I am wondering how would the performance change according to the number of grasp bins.
> >
> > ### Responses 2:
> >
> > According to the theory of antipodal grasping [2], the grasp/friction cone could tolerate a 15-degree offset for the friction factor of 0.3. In our settings, we use 5-degree discretization. Although the language-conditioned grasp requires higher precision than the general grasp, 36 bins with 5 degrees are enough. On the other hand, our model generates the continuous orientation distribution and could be sampled with any resolutions. However, more exploration in this direction is interesting but is not the focus of this paper.
> >
> > ### Questions 3:
> >
> > *In the writing, I hope the authors can state what's the input and output of the mapping that's equivariant to define the problem in a clearer way.*
> >
> > ### Response 3:
> >
> > Thanks for the feedback. We conduct convolution between language-conditioned steerable kernels and visual information encoded in feature maps. The convolution is equivariant in the sense that if we rotate any single object in the scene, the output object grasp pose will be rotated accordingly. We provide a visualization in Figure 1 and a local equivariance proof in Appendix A.2.2.
> >
> > ## References
> >
> > [1] Cohen, Taco S., and Max Welling. "Steerable CNNs." International Conference on Learning Representations. 2016.
> >
> > [2] Bicchi, Antonio. "On the closure properties of robotic grasping." The International Journal of Robotics Research 14.4 (1995): 319-334.

---

> ### Comment · Reviewer_n2g1 · 2023-11-23
>
> Thank you for the clarification.
>
> I have carefully read through the comments by peer reviewers as well as your rebuttal. It seems that there are still shared concerns about baselines among all reviewers. Your response 2 to my weakness 2 still seems insufficient to convince me without additional experiments.
>
> I'd maintain my score here.

---

### Official Review · Reviewer_vD3Z · 2023-11-06

**Soundness:** 3 good
**Presentation:** 2 fair
**Contribution:** 3 good
**Rating:** 6
**Confidence:** 3

**Summary:**

In this work, the authors propose a novel method to train language-conditioned grasping policies that are equivariant under the SE2 symmetry. To achieve this, the authors first divide the inputs into two encoders: One processes both language and RGB to obtain pixel-wise attention, and another part; Another one to expand language embedding to a steerable kernal which satisfies the equivariant constraints. Then the steerable kernal and the pixel-wise attention are "fused" together through cross correlation and inverse fourier transform to produce the action activations. The authors test the proposed method on diverse 2D pickup tasks and show that, their method can outperform naive baselines with only a few demonstrations.

**Strengths:**

1) The proposed method is quite novel as it leverages the symmetry that exists in robotic problems.
2) The method also demonstrate good 1-shot or few shot performances comparing with baselines.

**Weaknesses:**

1) It is rather strange to see that rotation is applied to the language embedding, but not the image space...Need an explanation on this.
2) Also, CLIP language features may not the best representation of actions such as "pick", "move" etc. Since the policy is grasping only it may not impose a bottlneck as of now.
3) Need some discussions on why the whole policy, not just the steerable kernel, is also equivariant under the SE2 symmetric group.
4) SE2 also includes a translation part, but I don't see that the steerable kernel is equivariant under translation.
5) In the paper it is not clear if there is a pre-training phase. It is hard to believe that the network weights (of \phi, and psi) are properly learned with just one trajectory if they are always randomly initialized from the beginning.

**Questions:**

N/A

---

> ### Author Response · Authors · 2023-11-21
>
> We express our gratitude to the reviewer for their valuable feedback on our manuscript. Here is our detailed response to the concerns raised:
>
> ## Weakness and Response
>
> ### Weakness 1:
>
> *It is rather strange to see that rotation is applied to the language embedding, but not the image space...Need an explanation on this.*
>
> ### Response 1:
>
> Excellent question. The direct way to apply rotation to image space is to feed a stack of rotated images to the network. It is computationally expensive compared with the use of a set of rotated kernels or steerable kernels, especially when the number of rotations is large. The language-conditioned steerable kernel is more manageable and efficient. Theoretically, we can derive from the **commutative** property of the convolution action $a*b=b*a$, where a, b denotes two feature maps and $*$ denotes convolution. Further, with group action $g \in SE(2)$, $(g\cdot a)*b = a*(g\cdot b)$.
>
> ### Weakness 2:
>
> *Also, CLIP language features may not the best representation of actions such as "pick", "move" etc. Since the policy is grasping only it may not impose a bottleneck as of now.*
>
> ### Response 2:
>
> Good observation. We used CLIP because its text encoder is trained to align with the image distribution, which makes it a motivation for some recent visual-text tasks in robotics [1][7]. However, as mentioned in section 3.2.2, our proposed framework is suitable to fit any pre-trained language encoder or VLM. Finding a better VLM model suitable for robotics tasks could be a potential future work. Moreover, since in this work the only robot action is “grasping”, if the language-conditioned kernel can learn to extract the object type from the CLIP text embedding, it is sufficient to perform correct grasping. Our experiments successfully demonstrate such learning ability of our language-conditioned kernel.
>
> ### Weakness 3:
>
> *Need some discussions on why the whole policy, not just the steerable kernel, is also equivariant under the SE2 symmetric group. SE2 also includes a translation part, but I don't see that the steerable kernel is equivariant under translation.*
>
> ### Response 3:
>
> Good question. The rotation symmetry is realized by the languaged-conditioned dynamic steerable kernel. The translational symmetry is realized by the cross-correlation action, which means that convolution actions are naturally equivariant with respect to translation before adding the additional SO(2)-equivariant property. Detailed explanations can be found in Appendix A.2.2.
>
> ### Weakness 4:
>
> *In the paper it is not clear if there is a pre-training phase. It is hard to believe that the network weights (of $\phi$, and $\psi$ ) are properly learned with just one trajectory if they are always randomly initialized from the beginning.*
>
> ### Responses 4:
>
> We do not engage in any pre-training. Our approach solely utilizes the pre-trained CLIP model. Specifically, for LEG-UNET, we utilize only the CLIP text encoder. In the case of LEG-Cliport, both the CLIP image encoder and text encoder are employed. Details regarding our training methodology and network specifications can be found in Section 4.3 and Appendix A.3, respectively.
>
> For few-shot learning, equivariant networks are noted for their fast convergence and enhanced sample efficiency. This has been substantiated by multiple studies, as referenced in [1], [2], [3], [4], [5], and [6].
>
> In our analysis, we focused on the symmetry inherent in the tasks and proposed a comprehensive framework that incorporates a dynamic steerable kernel. This framework is designed to exploit task symmetry, thereby achieving improved sample efficiency. We consider this ability to leverage symmetry as a significant strength of our work, rather than a limitation.
>
> ## References
>
> [1] Zhu, Xupeng, et al. "Sample Efficient Grasp Learning Using Equivariant Models." Robotics: Science and Systems. 2022.
>
> [2] Huang, Haojie, et al. "Equivariant Transporter Network." Robotics: Science and Systems. 2022.
>
> [3] Jia, Mingxi, et al. "Seil: Simulation-augmented equivariant imitation learning." 2023 IEEE International Conference on Robotics and Automation (ICRA). IEEE, 2023.
>
> [4] Wang, Dian, Robin Walters, and Robert Platt. "SO (2)-Equivariant Reinforcement Learning." International Conference on Learning Representations. 2022.
>
> [5] Wang, Dian, et al. "Equivariant $ q $ learning in spatial action spaces." Conference on Robot Learning. PMLR, 2022.
>
> [6] Yang, Jingyun, et al. "EquivAct: SIM (3)-Equivariant Visuomotor Policies beyond Rigid Object Manipulation." arXiv preprint arXiv:2310.16050 (2023).
>
> [7] Tang, Chao, et al. "Task-Oriented Grasp Prediction with Visual-Language Inputs." arXiv preprint arXiv:2302.14355 (2023).

---

### Official Review · Reviewer_C3bk · 2023-11-07

**Soundness:** 2 fair
**Presentation:** 2 fair
**Contribution:** 2 fair
**Rating:** 3
**Confidence:** 4

**Summary:**

The paper introduces a method called Language-conditioned Equivariant Grasp (LEG), which utilizes language instruction to guide robotic grasping. It presents a language-conditioned grasp network with a dynamic kernel generator to showcase the effectiveness of LEG on the Language-Grasp Benchmark that includes expert demonstrations. This benchmark comprises 10 language-conditioned grasping tasks in a simulated environment.

**Strengths:**

- The proposed method, Language-conditioned Equivariant Grasp (LEG), leverages the symmetries of language-conditioned robotic grasping by mapping the language instruction to a steerable kernel.
- The authors analyze the symmetry of language-conditioned grasp and propose a general framework to leverage it.
- The proposed method achieves high sample efficiency and grasping performance in both simulated and real-world robot experiments.

**Weaknesses:**

- The mainstream of grasping research is in SE(3), but this paper remains in SE(2).
This paper seems to have conducted implicit pose estimation or simply learned the rotation angles of objects. However, it appears to lack novelty.
- Real-world experiments are limited to a single task (pick-by-part) and a small set of objects. Some of the experimental objects are not sufficient to demonstrate effectiveness, as they are mostly cylindrical in shape. As long as the object is between the grippers, closing the grippers will definitely be able to grasp it.
- There are some obvious typos in the article, like the third line from the bottom of Section 4.

**Questions:**

Could you explain the reason for adding language embedding in the φ branch? It seems to be redundant and does not have significant impact on performance.

---

> ### Author Response · Authors · 2023-11-21
>
> We express our gratitude to the reviewer for their valuable feedback on our manuscript. Here is our detailed response to the concerns raised:
>
> ## Weakness and Response
>
> ### Weakness 1:
>
> *The mainstream of grasping research is in SE(3), but this paper remains in SE(2). This paper seems to have conducted implicit pose estimation or simply learned the rotation angles of objects. However, it appears to lack novelty.*
>
> ### Response 1:
>
> We respectfully disagreed with the reviewer that this is the weakness of our method.
>
> 1. The mainstream of SE(3) grasping does not indicate the lack of novelty of all SE(2) grasping currently. Language-conditioned grasping is being actively researched but still under-explored. As the other reviewers point out, there exist very recent publications such as [1][5] on SE(2) language-conditioned grasping. Most methods rely on segmentation [1, 2], cross-attention [5], or direct concatenation of language features and visual features [3, 4, 6]. To the best of our knowledge, our work is the first method that grounds language using language-condition steerable kernels, and we show its effectiveness compared with a strong baseline [6]. We are happy to address further concerns if the reviewer can kindly provide some related state-of-the-art work on language-conditioned grasping.
> 2. Our method, i.e., language-conditioned dynamic kernel, is straightforward to be extended to 3D to solve SE(3) grasping using 3D kernels, which is based on irreps of SO(3), i.e., Wigner-D matrices. This change is secondary compared to our approach on integrating language-conditioning.
> 3. To clarify, grasping is inherently a harder task compared to pose estimation. Previous work has demonstrated the complexity of the grasping problem [7, 8]. For example, a 6-Dof pose of a bowl does not necessarily provide information about where to grasp. And, object-level pose estimation methods cannot do our pick-part task.
>
> ### Weakness 2:
>
> *Real-world experiments are limited to a single task (pick-by-part) and a small set of objects. Some of the experimental objects are not sufficient to demonstrate effectiveness, as they are mostly cylindrical in shape. As long as the object is between the grippers, closing the grippers will definitely be able to grasp it.*
>
> ### Response 2:
>
> First, we chose the pick-part task for our real-world experiments because it represents a challenging yet intriguing aspect of robotic grasping. This task requires the model not only to pick the object but also to differentiate specific parts of it. And, we provide three metrics and experiments with different numbers of demonstrations to show the effectiveness of our method. We believe, given the difficulty of the task and various evaluations, our physical experiments are sufficient to demonstrate the effectiveness of the real-world applicability of our method.
>
> Second, we respectfully disagree with the cylindrical grasping argument. We have many objects that are not cylindrical. Take the bowl as an example; it is not in a cylindrical shape. The model needs to find the rim and reason the right angle of picking at a certain point on the rim. Meanwhile, simply grasping the estimated pose of the object will lead to failed grasping.
>
> Furthermore, beyond grasping, the model also needs to select the right object given language instruction. The model not only needs to pick the object but also needs to differentiate specific parts of it.

---

> > ### Author Response · Authors · 2023-11-21
> >
> > ## Questions and Response
> >
> > ### Questions 1:
> >
> > *Could you explain the reason for adding language embedding in the $\phi$ branch? It seems to be redundant and does not have a significant impact on performance.*
> >
> > ### Response 1:
> >
> > The main motivation is that we deliberately design the $\phi$ network as a vision-language module so that it provides flexibility for our framework to utilize any state-of-the-art VLMs. We provided visualizations in Figure 2 that $\phi$ network can highlight the region in the dense feature map from the observation. Also, empirically, $\phi$ network with language does not harm performance as shown in Table 4, meanwhile providing extra flexibility.
> >
> > Another motivation is that using language in both $\phi$ and $\kappa$ potentially enables multi-round reasoning between vision and language features, possibly allowing more complicated instructions.
> >
> > Besides, we have the ablation study in Table 4 to compare $\phi$ trained from scratch (LEG) and $\phi$ with an off-the-shelf pretrained VLM (LEG-Cliport). It shows that our method gains better performance by having the flexibility of using pre-trained VLM on $\phi$ .
> >
> > ## References
> >
> > [1] Tziafas, Georgios, et al. "Language-guided Robot Grasping: CLIP-based Referring Grasp Synthesis in Clutter." arXiv preprint arXiv:2311.05779 (2023).
> >
> > [2] Rashid, Adam, et al. "Language embedded radiance fields for zero-shot task-oriented grasping." 7th Annual Conference on Robot Learning. 2023.
> >
> > [3] Chen, Yiye, et al. "A joint network for grasp detection conditioned on natural language commands." 2021 IEEE International Conference on Robotics and Automation (ICRA). IEEE, 2021.
> >
> > [4] Shao, Lin, et al. "Concept2robot: Learning manipulation concepts from instructions and human demonstrations." The International Journal of Robotics Research 40.12-14 (2021): 1419-1434.
> >
> > [5] Tang, Chao, et al. "Task-Oriented Grasp Prediction with Visual-Language Inputs." arXiv preprint arXiv:2302.14355 (2023).
> >
> > [6] Shridhar, Mohit, Lucas Manuelli, and Dieter Fox. "Cliport: What and where pathways for robotic manipulation." Conference on Robot Learning. PMLR, 2022.
> >
> > [7] Huang, Haojie, et al. "Edge grasp network: A graph-based se (3)-invariant approach to grasp detection." *2023 IEEE International Conference on Robotics and Automation (ICRA)*. IEEE, 2023.
> >
> > [8] Mahler, Jeffrey, et al. "Learning ambidextrous robot grasping policies." *Science Robotics* 4.26 (2019): eaau4984.

---

### Official Review · Reviewer_FTbR · 2023-11-10

**Soundness:** 3 good
**Presentation:** 3 good
**Contribution:** 2 fair
**Rating:** 5
**Confidence:** 3

**Summary:**

The paper aims to tackle the problem of 2D grasping conditioned on language instructions. To be more specific, the language instructions are provided as grasping an object by the name, color or a specific part and they use behavior cloning to learn the grasping policy from language annotated demonstrations. Moreover, in order to incorporate language into the grasping policy in a way that exploits the symmetry and geometry in the task, the authors proposed a framework that maps the language instruction to a SO(2) steerable kernel. They demonstrate the effectiveness of their method on their proposed grasping benchmark in the simulation and on the real robot.

**Strengths:**

The author proposed a novel dynamic kernel that maps the language instructions and has provided theoretical guarantee for the rotation symmetry.  The paper also shows its strength of high sample efficiency and better grasping performance both in simulation and on real world. The author also proposed a grasping simulation benchmark annotated with language instructions that can be used for future work on 2d grasping.

**Weaknesses:**

The author fails to compare their method against some most recent work in language conditioned grasping. For example, this work[1] also has the task of semantic grasping(grasping some object by some particular parts). Moreover, this paper only focuses on 2D grasping with language instructions but some objects may not be graspable from just a 2d top-down grasp. Some work[2] have also shown their method on 4D grasping in cluttered scene with language instructions.

In the real world experiment, the tested objects and variations are very limited compared to the simulation. It is more convincing to show more objects in the real world experiments and potentially show the results of the different methods not just LEG-Unet in the real world.

[1]Sundaresan, P., Belkhale, S., Sadigh, D., & Bohg, J. (2023). Kite: Keypoint-conditioned policies for semantic manipulation. arXiv preprint arXiv:2306.16605.

[2]Tziafas, G., Yucheng, X. U., Goel, A., Kasaei, M., Li, Z., & Kasaei, H. (2023, August). Language-guided Robot Grasping: CLIP-based Referring Grasp Synthesis in Clutter. In 7th Annual Conference on Robot Learning.

**Questions:**

1. The author proposed a grasping benchmark in pybullet simulation with objects imported from YCB and GraspNet Dataset with language annotations. However, the paper doesn't compare the diversity and the complexity of the grasping objects in this grasping benchmark  with other grasping benchmark commonly used by the robotic grasping research. It would be more convinced to have a table with comparisons with other grasping dataset/benchmarks.
2. The current language instructions used in the experiments are provided in the appendix. I am wondering if this method can generalize to other language instructions that have the similar meaning but don't exist in the training dataset or the language instructions that are in the free-form.
3. For the experiment section, the authors mention that LEG-Unet is better than LEG-CLIPort on most tasks. Could the authors provide more explanation for this result?

---

> ### Author Response · Authors · 2023-11-21
>
> We express our gratitude to the reviewer for their valuable feedback on our manuscript. Here is our detailed response to the concerns raised:
>
> ## **Weakness and Response**
>
> ### Weakness 1:
>
> *The author fails to compare their method against some most recent work in language conditioned grasping. For example, this work[1] also has the task of semantic grasping(grasping some object by some particular parts). Moreover, this paper only focuses on 2D grasping with language instructions but some objects may not be graspable from just a 2d top-down grasp. Some work[2] have also shown their method on 4D grasping in cluttered scene with language instructions.*
>
> ### Response 1:
>
> - We appreciate the suggestion and have updated the related work section to include a comprehensive comparison with all relevant studies known to us.
> - We clarify that KITE[1] is a 3D policy network, primarily focusing on aspects other than grasping. The key pixel identification mechanism in KITE aligns closely with the attention module of Cliport[3], which serves as our primary baseline. The major difference between KITE and our Cliport-UNet is KITE's use of depth images projected into point clouds for learning in 3D space, whereas our Cliport-UNet decodes pixel features into 2D grasp orientations. Additionally, we provide some key differences between our method and KITE[1] as shown in **Table** R1.
> - We acknowledge the recent work OCID-VLG/CROG[2], which shares a similar focus. We've thoroughly analyzed the differences between our method and theirs in the updated section. We also note that this is a very recent publication lacking available code. Moreover, the 4D grasp they refer to involves top-down grasping with gripper width prediction, differing from 6 DoF grasping in 3D space.
>
> ### Weakness 2:
>
> *In the real world experiment, the tested objects and variations are very limited compared to the simulation. It is more convincing to show more objects in the real world experiments and potentially show the results of the different methods not just LEG-Unet in the real world.*
>
> ### Response 2:
>
> - The critique about the limited variety of objects and variations in our real-world experiments compared to the simulations has been noted. We chose the pick-part task for our real-world experiments because it represents a challenging yet intriguing aspect of robotic grasping. This task requires the model not only to pick the object but also to differentiate specific parts of it.
> - We evaluated performance using three metrics: grasping success rate, accuracy in selecting the correct object, and precision in grasping the intended part. The model was tested with varying numbers of demonstrations (1,5) and on unseen objects. The consistency of our real-world experiment's performance with the simulated pick-part experiments was established.
> - Additionally, our manuscript's Tables 1 and 2 compare our method's performance with baselines across 13 tasks involving hundreds of objects. Thus, we believe our physical experiment sufficiently demonstrates the real-world applicability of our method.

---

> > ### Author Response · Authors · 2023-11-21
> >
> > ## **Questions and Responses**
> >
> > ### Question 1:
> >
> > *The author proposed a grasping benchmark in pybullet simulation with objects imported from YCB and GraspNet Dataset with language annotations. However, the paper doesn't compare the diversity and the complexity of the grasping objects in this grasping benchmark with other grasping benchmark commonly used by the robotic grasping research. It would be more convinced to have a table with comparisons with other grasping dataset/benchmarks.*
> >
> > ### Response 1:
> >
> > - **Benchmark Comparison:** We provide a comparative analysis through Table R1 in our manuscript. Our LEG-benchmark, as detailed in Section 4.1.1 of our paper, is unique as it's the first language-conditioned grasping benchmark with an open-source framework and fine-grained tasks, to the best of our knowledge. Table R1 contrasts our benchmark with others, focusing on the framework, simulator, grasping oracle functions, and fine-grained tasks that emphasize evaluating a model's semantic-geometric compositional ability. In contrast, benchmarks like OCID-VLG offer a large-scale dataset for cluttered scenes, aimed at enhancing grasping generalization.
> >
> >
> >     | Table R1 | Open-source simulator | Simulation oracles | Language instruction | Fine-grained tasks | Action |
> >     | --- | --- | --- | --- | --- | --- |
> >     | GraspNet[4] | ✅ | ✅ | ❌ | ❌ | 6-DoF |
> >     | OCID-VLG[2] | ❌ | ❌ | ✅ | ❌ | 4-DoF |
> >     | Ours | ✅ | ✅ | ✅ | ✅ | 4-DoF |
> > - **Algorithm Comparison:** In Table R2, we draw comparisons with CLIPort, given the close similarity in problem settings.
> >
> >
> >     | Table R2 | Language-conditioned | Few-shot learning | Extra-labeling-free | Part grasping | End-to-end | Action |
> >     | --- | --- | --- | --- | --- | --- | --- |
> >     | KITE[1] | ✅ | ✅ | ❌ (Keypoints) | ✅ | ❌ | 6-DoF |
> >     | CROG[2] | ✅ | ❌ | ❌ (Seg mask) | ❌ | ✅ | 4-DoF |
> >     | CLIPort[3] | ✅ | ✅ | ✅ | ❌ | ✅ | 4-DoF |
> >     | Ours | ✅ | ✅ | ✅ | ✅ | ✅ | 4-DoF |
> >
> > ### Question 2:
> >
> > *The current language instructions used in the experiments are provided in the appendix. I am wondering if this method can generalize to other language instructions that have the similar meaning but don't exist in the training dataset or the language instructions that are in the free-form.*
> >
> > ### Response 2:
> >
> > - **Generalization Capability:** Addressing this important aspect, we provided results of LEG-UNETL trained with a specific language instruction and tested with unseen, similar language instructions. The findings, presented in our manuscript's Table, demonstrate a slight decrease but affirm the model's capability to interpret and respond to flexible language instructions. Further, we here provide an extended experiment as shown in Table R3, where we train the model with one fixed language template and test the model with unseen instructions, given 1/10 demonstrations. The results show that our model can generalize to *free-form* language instructions with similar performance as reported in our manuscript.
> >
> >
> >     | Table R2 | pick-parts | pick-random-v1 | pick-random-v2 |
> >     | --- | --- | --- | --- |
> >     | 1 demo | 88.25% | 88.6% | 53.4% |
> >     | 10 demos  | 95.25% | 94.6% | 85.6%  |
> >
> > ### Question 3:
> >
> > *For the experiment section, the authors mention that LEG-Unet is better than LEG-CLIPort on most tasks. Could the authors provide more explanation for this result?*
> >
> > ### Response 3:
> >
> > - **Performance Analysis:** We clarify that LEG-UNet, especially when trained from scratch, shows superior performance over LEG-Cliport on tasks that primarily involve shape differences. This could be attributed to two factors:
> >     1. The data preprocessing in LEG-Cliport to reshape images for feeding into the CLIP image encoder, which might impact the CLIP geometric features of each pixel.
> >     2. Training from scratch likely retains more of the original geometric information, providing an edge in performance.
> >
> > ## References
> >
> > [1] Sundaresan, Priya, et al. ”Kite: Keypoint-conditioned policies for semantic manipulation.” arXiv preprint arXiv:2306.16605 (2023).
> >
> > [2] Tziafas, Georgios, et al. ”Language-guided Robot Grasping: CLIP-based Referring Grasp Synthesis in Clutter.” 7th Annual Conference on Robot Learning. 2023.
> >
> > [3] Shridhar, Mohit, Lucas Manuelli, and Dieter Fox. ”Cliport: What and where pathways for robotic manipulation.” Conference on Robot Learning. PMLR, 2022.
> >
> > [4] Fang, Hao-Shu, et al. ”Graspnet-1billion: A large-scale benchmark for general object grasping.” Proceedings of the IEEE/CVF conference on computer vision and pattern recognition. 2020.

---

### Author Response · Authors · 2023-11-22

# General Response to Reviewers

We express our gratitude to the reviewers for their insightful feedback on our manuscript. We are happy to note that the majority of reviewers have acknowledged the novelty and the superior performance of our proposed method compared to existing baselines.

## Reviewer Feedback

### Reviewer FTbR

- **Comment:** "The authors have introduced an innovative dynamic kernel that effectively maps language instructions, accompanied by a theoretical guarantee for rotation symmetry. The paper impressively demonstrates high sample efficiency and enhanced grasping performance, both in simulations and in real-world applications."

### Reviewer C3bk

- **Comment:** "The method proposed achieves remarkable sample efficiency and grasping performance in both simulated and real-world robotic experiments."

### Reviewer vD3Z

- **Comment:** "The method is quite innovative, capitalizing on the inherent symmetry in robotic challenges. It also exhibits commendable one-shot or few-shot performance, surpassing various baselines."

### Reviewer n2g1

- **Comment:** "The evaluation is robust, incorporating a range of baselines and environments, and notably includes real-world grasping experiments with unknown objects."

### Reviewer fe34

- **Comment:** "The methods proposed demonstrate strong grasping performances in the established benchmark."

## Clarity of Presentation

- **Reviewer n2g1 on Clarity:** "The writing is clear and easy to understand."

## Manuscript Modifications

In response, we have updated our manuscript, highlighted in blue, to enhance our comparison with related work and to elaborate on the experiments involving unseen, similar language instructions. We have also addressed the concerns raised by individual reviewers in a point-by-point manner.

---

### Meta-Review · Area_Chair_VTGH · 2023-12-18

**Metareview:**

The authors introduce a steerable kernel for collapsing symmetries in the pose space as defined by the language. The intuition is straightforward and the performance (particularly given low sample set) noticeably improves over baselines like CLIPort.  The space is rather crowded for relevant papers to discuss that make different decisions in their grasping pipeline, what to condition on, or the output space. This paper falls right on the borderline of the acceptance for the group.

**Justification For Why Not Higher Score:**

The results are not as comprehensive as other papers in this space and the work not as well articulated.

**Justification For Why Not Lower Score:**

The experiments are promising and the task intuitive.

---

### Decision · Program_Chairs · 2024-01-16

Reject